# THE LOSS KERNEL: A GEOMETRIC PROBE FOR DEEP LEARNING INTERPRETABILITY

## ABSTRACT

We introduce the loss kernel, an interpretability method for measuring similarity between data points according to a trained neural network. The kernel is the covariance matrix of per-sample losses computed under a distribution of low-loss-preserving parameter perturbations. We first validate our method on a synthetic multitask problem, showing it separates inputs by task as predicted by theory. We then apply this kernel to Inception-v1 to visualize the structure of ImageNet, and we show that the kernel's structure aligns with the WordNet semantic hierarchy. This establishes the loss kernel as a practical tool for interpretability and data attribution.

## 1 INTRODUCTION

A central goal in AI interpretability and data attribution is interpreting and mapping the global structure of the data distribution as seen by a trained neural network (Carter et al., 2019; Pepin Lehalleur et al., 2025; Olah, 2015). One approach is to start local, by quantifying a suitable measure of similarity between pairs of individual samples — that is, by defining a kernel. "Interpreting" the global structure of the data distribution then becomes a problem of analyzing the geometric structure in this kernel (e.g., via clustering techniques), and "mapping" becomes a problem of visualizing points in this kernel space (e.g., via dimensionality reduction techniques).

This kernel-based approach has been used successfully with similarity measures derived from activations or representations. For example, it is possible to define a kernel via cosine similarity between the hidden vectors of sparse autoencoders (SAEs). Applying UMAP to this kernel provides a way to visualize the space of features in language models (Bricken et al., 2023; Templeton et al., 2024) and image models (Gorton, 2024). This kernel has also been used for analysis, such as to determine the (nearly) hierarchical relations between features (Bricken et al., 2023).

In this paper, we take a different approach derived from the geometric structure of the loss landscape. Neural networks are *singular* models, meaning many different parameter vectors encode identical functions and achieve the same loss. Rather than studying individual weight settings, singular learning theory (SLT; Watanabe 2009), which studies these singular models, suggests analyzing the entire set of low-loss solutions. This perspective motivates us to define the **loss kernel**, a measure of functional similarity based on shared sensitivity to parameter perturbations restricted to this low-loss set of solutions. Formally, the loss kernel, $K(x, x')$, is given by the covariance matrix of per-sample losses, $\mathrm{Cov}[\ell(x; w), \ell(x'; w)]$, under perturbations drawn from a suitable probe distribution. A high covariance value indicates that the inputs $x$ and $x'$ share sensitivity to the same parameter perturbations, which provides evidence for two samples being functionally coupled inside a given model.

We demonstrate the loss kernel as a practical interpretability technique by combining it with established kernel-based techniques to study two settings. First, in a controlled experiment using a synthetic multitask arithmetic problem, we confirm that the kernel successfully separates inputs corresponding to functionally independent subtasks, as predicted by theory. Second, we apply the loss kernel to an Inception-v1 model to create a visual map (Figure 1) of the ImageNet dataset on which it was trained (Szegedy et al., 2014; Deng et al., 2009). We then quantitatively validate that the structure of this kernel reveals a coherent semantic organization that is consistent with the WordNet class taxonomy (Princeton University, 2010).

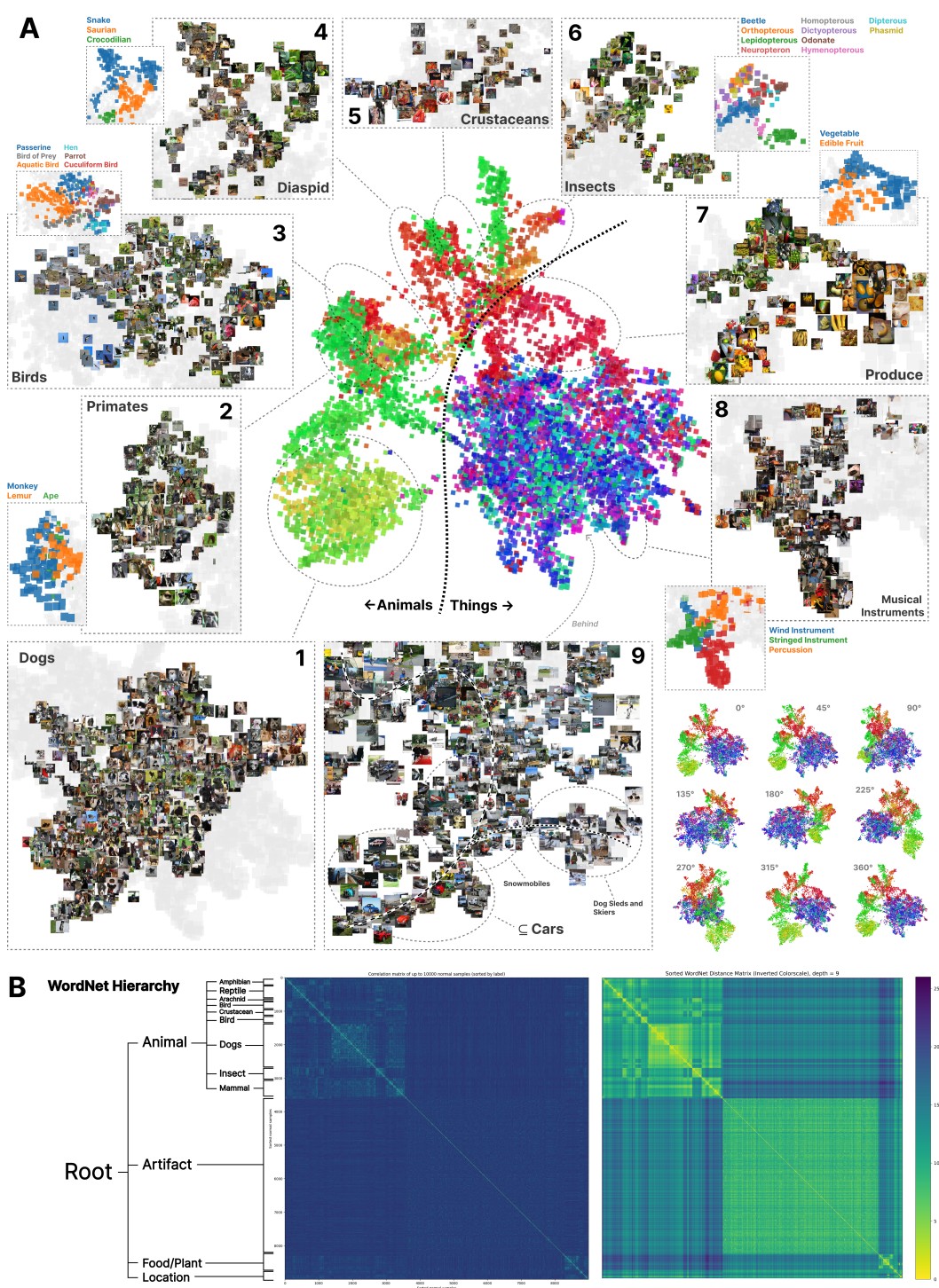

**Figure 1: Geometry of the Loss Kernel for InceptionV1 on ImageNet. A** UMAP of pairwise distances induced by the normalized loss kernel $R(x, x') = \text{Corr}_{w \sim p(w|\mathcal{D})}[\ell(x; w), \ell(x'; w)]$ for InceptionV1 on ImageNet-1k; each point is one image, colored continuously by position in the ImageNet hierarchy. Similar colors indicate inputs are semantically similar. **1–9** Insets: example neighborhoods with thumbnails showing coherent regions for *dogs* (1), *primates* (2), *birds* (3), *diaspids* (4), *crustaceans* (5), *insects* (6), *produce* (7), *musical instruments* (8), and *vehicles/cars* (9). **Bottom right** Orbit views of the same 3-D embedding. **B** The full correlation kernel matrix (10k×10k) next to the ground truth distance matrix derived from the ImageNet hierarchy shows similar block structures in both.

**Contributions.** Our contributions are thus:

- **We introduce the loss kernel as a measure of functional coupling**, motivating it from the geometric perspective of singular learning theory and defining it through a principled, local probe distribution. (Section 2)

- **We validate the loss kernel in a controlled setting**, confirming that the loss kernel is able to successfully separate subtasks in a synthetic multitask experiment, as predicted theoretically. (Section 3)

- **We apply the loss kernel to Inception-v1 on ImageNet**, demonstrating its utility as a large-scale interpretability and visualization tool. We show that its structure reveals a coherent semantic organization consistent with the WordNet class taxonomy. (Section 4)

## 2 THE LOSS KERNEL

In this section, we define the loss kernel, a metric that quantifies whether two inputs are processed similarly by a trained neural network. First (Section 2.1), we motivate our focus on the geometry of the loss landscape, specifically the set of low-loss points $W_\epsilon$ that contains a given trained model $w^*$. Second (Section 2.2), we develop a practical *probe distribution* using a localized Gibbs posterior, which allows us to sample from this low-loss region. Finally (Section 2.3), using this distribution, we formally define the *loss kernel* as the covariance of per-sample losses under our probe distribution.

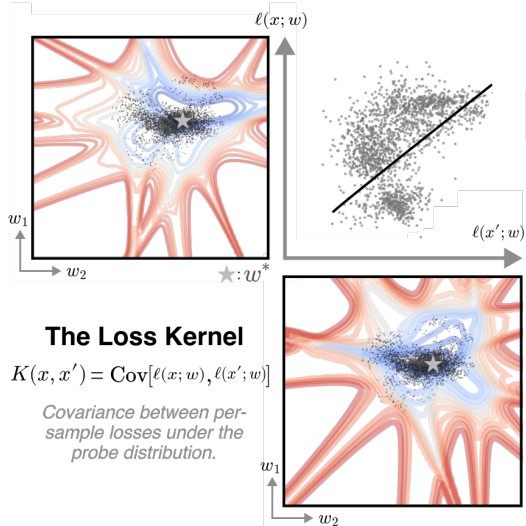

**The Loss Kernel**

$$K(x, x') = \mathrm{Cov}[\ell(x; w), \ell(x'; w)]$$

*Covariance between per-sample losses under the probe distribution.*

### 2.1 INTERPRETABILITY AND DEGENERACY

The typical process of training a neural network yields a single parameter vector $w^*$, optimized via an algorithm like SGD against an objective function of the form

$$L_n(w) = \sum_{i=0}^{n} \ell(x_i; w)$$

where $\ell(x_i; w)$ is the loss on $i$-th data sample $x_i$ for the parameter vector $w$, with a dataset of size $n$.

The field of *interpretability* seeks to understand the structure of the trained model represented

**Figure 2: The Loss Kernel.** The loss kernel $K(x, x')$ is the covariance of per-sample losses $\ell(x, w)$ for two inputs $x$ and $x'$, computed over a *probe distribution* of model weights $w$ (gray points) sampled near a trained solution $w^*$. These two losses respond differently to different weights (top left, bottom right), reflecting which parts of the model are important for those inputs. A positive correlation in these losses (scatter plot, top right) signifies that the two inputs share sensitivity to the same weight perturbations, which we interpret as evidence that the model is treating the inputs $x$ and $x'$ similarly.

by $w^*$. It is typically implicitly presumed that one can understand the structure in the trained model using the parameters $w^*$, either directly by inspecting, for example, weight magnitudes (Kovaleva et al., 2021), or indirectly by examining the computation process of the model at $w^*$ through, for example, activations (Bricken et al., 2023; Wang et al., 2022; Carter et al., 2019) and gradients (Ancona et al., 2018; Sundararajan et al., 2017).

A challenge to interpreting weights directly is that neural networks are *singular*: many different parameters implement the same function or achieve the same loss. This degeneracy means that properties specific to $w^*$ may reflect arbitrary details of the learned implementation that are irrelevant to downstream behavior. For example, ReLU-scaling symmetries mean the absolute magnitude of individual weights or gradients is not always meaningful on its own, which undermines interpretability methods that rely on it.

**Singular Learning Theory.** Watanabe's (2009) singular learning theory (SLT) provides a mathematical framework for studying models that exhibit such degeneracies. A key idea from SLT is to study the geometry of the set of minima of the loss function as a whole rather than individual weight settings. Consider the set of parameters which are "almost equivalent" to $w^*$, according to the training loss $L_n(w)$:[1]

$$W_\epsilon = \{w \in \mathbb{R}^d \mid L_n(w) - L_n(w^*) < \epsilon\}. \tag{1}$$

The asymptotic volume-scaling behavior of (the population version of) $W_\epsilon$ is directly linked, through SLT, to the complexity, description length, and generalization error of the model at $w^*$ (Lau et al., 2025; Urdshals et al., 2025). Our work builds on this premise to develop a principled technique for measuring whether two inputs are processed in similar ways by a given trained neural network.

## 2.2 CONSTRUCTING A PRACTICAL PROBE

While $W_\epsilon$ is theoretically natural, it is difficult to integrate over this set because it is so high-dimensional. Moreover, we need a way to localize this set to a specific set of model weights obtained via stochastic optimization. We make two modifications to overcome these challenges and develop a practical low-loss probe:

**From Hard to Soft Constraints.** First, we replace the sharp boundary of $W_\epsilon$ with a smooth Gibbs factor, $\exp(-\beta L_n(w))$. This concentrates sampling in low-loss regions, where the inverse temperature $\beta$ plays a role analogous to $1/\epsilon$. This makes the distribution amenable to gradient-based MCMC sampling and is formally justified by the relationship between integrals over low-loss sets and expectations under the Gibbs distribution (see Appendix A.3).

**From Global to Local.** Second, we focus on the neighborhood containing the specific model $w^*$ found by a given run of stochastic optimization. The global loss landscape may contain many regions of low loss, but we wish to interpret the particular solution our training procedure has found. We therefore re-weight the Gibbs distribution with a Gaussian kernel centered at $w^*$.

This yields the final probe distribution over the training set $\mathcal{D}$:

$$p(w|\mathcal{D}) \propto \underbrace{\exp(-\beta L_n(w))}_{\text{Low-Loss Constraint}} \cdot \underbrace{\mathcal{N}(w|w^*, \gamma^{-1}I)}_{\text{Locality Constraint}}.$$

From a Bayesian perspective, this is equivalent to a *tempered Bayesian posterior* with Gaussian prior.

## 2.3 THE LOSS KERNEL

**The Loss Kernel.** The loss kernel, $K$, is the covariance matrix of per-sample losses under our probe distribution:

$$K(x, x') = \text{Cov}_{w \sim p(w|\mathcal{D})} \left[\ell(x; w), \ell(x'; w)\right]. \tag{2}$$

A high value of $K(x, x')$ indicates that inputs $x$ and $x'$ are functionally coupled, sharing sensitivity to the same parameter perturbations. The kernel is symmetric positive semi-definite as it is a covariance kernel. For analysis and visualization, we often use its normalized form:

$$R(x, x') = \frac{K(x, x')}{\sqrt{K(x, x)K(x', x')}},$$

with $R(x, x') = 0$ if $K(x, x) = 0$ or $K(x', x') = 0$, which measures the *correlation* between per-sample losses. $R(x, x')$ also has the advantage of being invariant under affine changes of the loss function, unlike $K(x, x')$ itself.

---

[1]Throughout the paper, we define objects using the training loss $L_n(w)$, including the loss kernel itself. Alternatively, we could define these objects using the population loss $L(w) = \lim_{n \to \infty} L_n(w)$, and treat the empirical versions as *estimators* of the population versions. We explore this further in Appendix A.1.4.

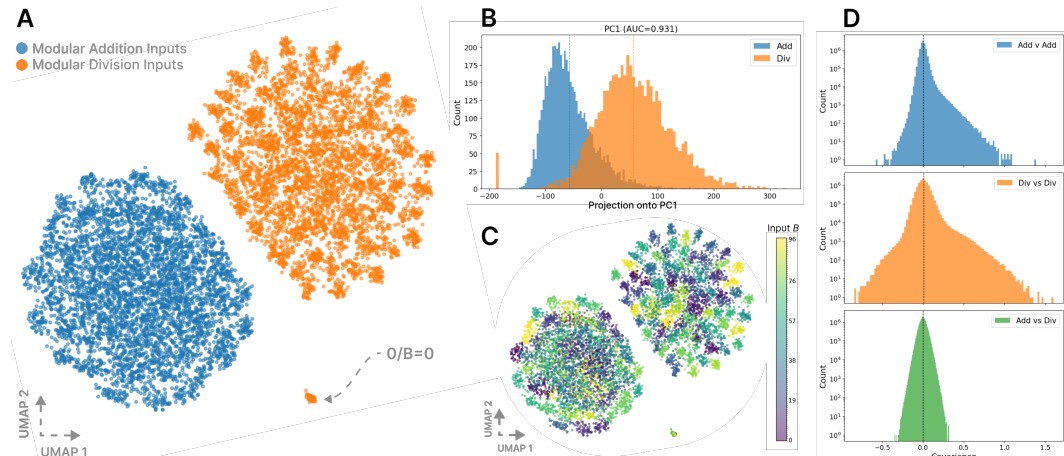

**Figure 3: Geometry of the loss kernel for a multitask modular-arithmetic model** ($p = 97$). (**A**) UMAP of pairwise distances derived from the loss kernel ($d(x, x') = 1 - R(x, x')$. Two well-separated clusters correspond to modular addition (blue) and modular division (orange). A small satellite cluster corresponds to the trivial modular division case $a = 0$, for which $0/b \equiv 0$ (mod 97). (**B**) Distribution of projections onto the first principal component of the normalized per-sample expected loss vectors, $\mathbb{E}[\ell(x_i; w)] - \ell(x_i; w^*)$. A single axis suffices to separate tasks (ROC–AUC = 0.931). (**C**) Same UMAP as in (**A**), colored by the value of input $b$. (**D**) Log-scaled covariance distributions for Addition vs. Addition, Division vs. Division, and Addition vs. Division pairs. Within-task covariances are heavy-tailed and skewed, whereas cross-task covariances are narrowly concentrated and approximately normal.

**Interpretation.** The loss kernel can be seen as a generalized version of the negated (local) Bayesian Influence Function (Kreer et al., 2025), which itself generalizes the influence function from classical statistics, see Appendix A.2. The diagonal of this kernel, $K(x, x)$, is the per-sample loss variance. Up to a multiplicative constant, the sum of $K(x, x')$ over the training set, $\sum_i K(x_i, x_i)$, is an empirical estimator for the *singular fluctuation*, a key quantity in SLT that governs the model's (Gibbs) generalization error, see Appendix A.1.

**Practical Estimation.** Expectations over the probe distribution $p(w|\mathcal{D})$ are intractable to compute analytically. We therefore approximate them using Monte Carlo methods. Specifically, we generate a set of $S$ samples $\{w_s\}_{s=1}^S$ from a Stochastic Gradient Langevin Dynamics (SGLD; Welling & Teh 2011) chain (or multiple parallel chains) initialized at the trained model's parameters, $w^*$. We then use these samples to compute standard unbiased plug-in estimators for the loss kernel $\hat{K}(x, x')$ and its normalized version $\hat{R}(x, x')$. We provide further details and departures from SGLD in Appendix B.

## 3 VALIDATION ON A SYNTHETIC TASK

Before using the loss kernel to explore structure in natural data, we first verify that it behaves as expected in a controlled scenario. Theoretically, we expect that for tasks solved by *independent* mechanisms — where the loss factorizes into a sum of sublosses depending on disjoint sets of weights — the cross-task loss covariance is *zero* (Appendix A.4). We test this prediction on a transformer trained on a multitask modular arithmetic problem designed to encourage such independent mechanisms.

**Multitask Arithmetic.** For our controlled scenario, we analyze a two-layer transformer on a multitask modular arithmetic ("grokking") problem, extending the single-task setup of Power et al. (2022). Our model is trained to perfect accuracy on two independent tasks: modular addition and modular division, both modulo 97. To encourage the development of distinct computational pathways, each operation uses a separate input vocabulary.

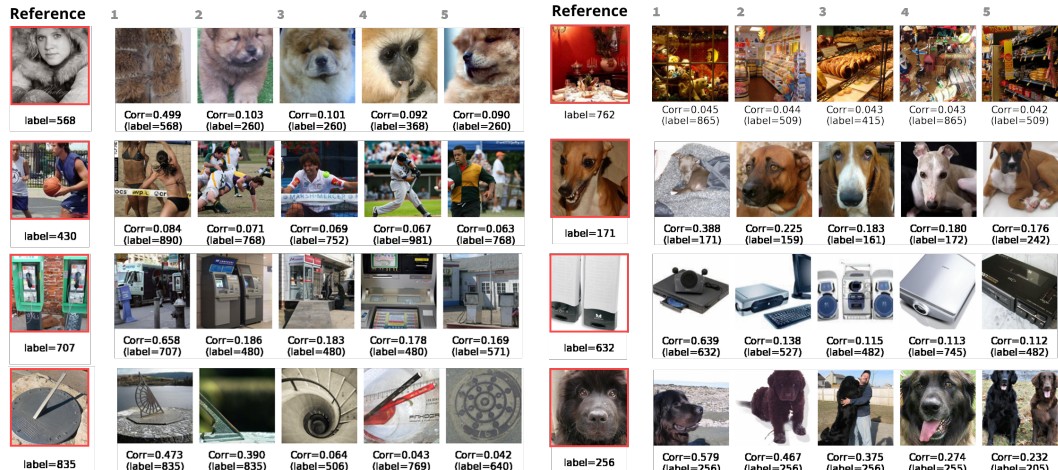

**Figure 4: Top-correlated examples under the loss kernel reveal interpretable patterns.** For each reference image (leftmost column), we show the top five most-correlated inputs under the loss correlation kernel $R$. We observe clustering by texture (e.g., fluffy fur coat and fluffy animals), shape (e.g., circular objects and line angle), color and category (e.g., people playing sports, electronics on a white background, dark vs. light brown dogs), and spatial layout (e.g., cluttered rooms). Additional visualizations are provided in Appendix D.7, and further computed correlation results are available at https://github.com/singfluence-anon/sf_imagenet_corrs

**Reducing Dimensionality.** We visualize the kernel by applying standard dimensionality reduction techniques to a set of reference points in the kernel space. We use UMAP, which obtains a low-dimensional embedding optimized to preserve nearest-neighbor relationships (McInnes et al., 2018).

UMAP operates on a distance matrix, where a point must have distance 0 with itself and positive distance with all other points. We transform the normalized loss kernel, or correlation, $R$ into a distance $d$ by setting the distance between any two samples $x$ and $x'$ to $d(x, x') = 1 - R(x, x')$. Applying UMAP to these pairwise distances produces the embedding depicted in Figure 3, where proximity in the visualization indicates a strong functional coupling between samples as measured by the kernel.[2]

**Interpreting the Kernel.** After computing the loss kernel over all pairs over 10,000 inputs drawn equally from both tasks, we find its structure reflects the task-level separation between addition and division. As seen in the UMAP visualization in Figure 3, the kernel separates into two distinct clusters corresponding precisely to the addition and division samples (and a third smaller cluster for the trivial modular division case where the dividend is zero). Examining the underlying covariance values confirms this observation: cross-task covariances are narrowly distributed around zero, while within-task covariances are substantially larger.

Though we lack a sufficient mechanistic understanding to establish whether this model's internal implementations of modular addition and division satisfy the criteria in Appendix A.4, observing vanishing correlation between tasks is consistent with the behavior theoretically predicted for functionally disjoint mechanisms. This establishes the kernel's utility in a setting with partially known ground-truth structure.

## 4 APPLICATION TO IMAGENET

Having established theoretically and empirically that the loss kernel can identify ground-truth functional separation in a controlled setting, we now deploy it as an exploratory tool on a large-scale, real-world task. We consider an Inception-v1 model (Szegedy et al., 2014) trained on ImageNet

---

[2]In the ImageNet setting we remove connections between inputs of the same label during UMAPs nearest neighbor search to eliminate potential spurious correlations (see Appendix B)

data (Deng et al., 2009), where the true functional organization is not fully known. Our goal is to investigate qualitatively whether we can use the kernel as a visualization tool and quantitatively whether structure in the kernel corresponds to meaningful semantic and hierarchical structure in the data.

**Visualizing the Loss Kernel.**    For 10,000 random validation examples, we compute the loss correlation matrix and examine top-correlated inputs. We find that nearest neighbors are interpretable, often sharing patterns of color, texture, shape, or content. Figure 4 provides qualitative examples of these relationships, showing the top and bottom correlated examples for a selection of inputs. Additional randomly chosen examples are available in Appendix D.7.

**Hierarchical Structure in ImageNet.**    The ImageNet dataset (Deng et al., 2009) is not a flat collection of classes; its labels are drawn from and organized according to the WordNet hierarchy, a large lexical database of English where nouns, verbs, adjectives, and adverbs are grouped into sets of synonyms (synsets), each expressing a distinct concept (Princeton University, 2010). Each node in the ImageNet hierarchy represents a category (e.g., "animals", "mammals", "devices", "plants"), and each leaf node corresponds to a specific class, which the model was trained to predict (e.g., "wire-haired fox terrier , "goldfish", "castle"). This taxonomy provides a natural (though only partial) source of ground truth for establishing similarity between ImageNet inputs, based on the similarity between their output labels according to the WordNet hierarchy.

To visualize this ground-truth structure overlaid on the loss kernel, we color each sample in Figure 1 (A) by the position of that sample's label in the ImageNet hierarchy. The version of ImageNet we use in these experiments is organized into 1,000 classes; by sorting these classes via their position in the hierarchy we assign similar hues to inputs of nearby categories.

**Hierarchical Structure in the Loss Kernel.**    The UMAP visualization in Figure 1 reveals a clear high-level organization that mirrors the primary branches of the WordNet hierarchy. A prominent split separates "animals" from "things," with a transitional region occupied by "produce" (Inset 7). Within these broad domains, the kernel captures finer taxonomic distinctions. For example, the "animal" kingdom subdivides into coherent superclasses. A large cluster representing "domesticated animals", particularly "dogs" (Inset 1), transitions into other mammals like "primates" (Inset 2), and then to "birds" (Inset 3). Nearby, we observe distinct groupings for "diapsids" (Inset 4), "crustaceans" (Inset 5), and "insects" (Inset 6). This hierarchical organization persists at deeper levels of specificity, as shown by the more detailed insets for "musical instruments" (Inset 8). The block structure of the full correlation matrix, when sorted by the WordNet hierarchy (Figure 1 B), provides an additional confirmation of this nested structure, showing strong intra-class correlation that closely mirrors the ground-truth semantic distance matrix derived from WordNet.

**The Kernel as a Developmental Tool.**    At initialization the kernel shows no coherent structure (see Figure 8). As training proceeds, structure begins to emerge. Early checkpoints separate broad regimes (e.g., "animal" vs. "thing"), mid-training checkpoints resolve salient subgroups (e.g., "dogs" forming a distinct cluster), and later checkpoints exhibit finer-grained specialization. The UMAP snapshots in Figure 8 illustrate this coarse-to-fine trajectory, where neighborhoods that are initially mixed become progressively more taxonomically coherent as training converges.

## 5 RELATED WORKS

**Bayesian Influence Functions and Training Data Attribution.**    The loss kernel we propose is a generalization of the negative (local) Bayesian Influence Function (BIF; Kreer et al. 2025), which has its roots in Bayesian sensitivity analysis (Giordano et al., 2017; Iba, 2025). (Kreer et al., 2025) introduced the BIF as a tool for Training Data Attribution (TDA; Koh & Liang 2020), a task focused on *provenance*—identifying which training points are most responsible for a specific model behavior. Our work addresses a different question: one of *functional coupling*. We generalize the BIF from a unidirectional, single-point attribution measure into a global, symmetric, positive semidefinite kernel that measures the functional relationship between arbitrary pairs of inputs. Furthermore, we are the first to demonstrate its power for large-scale interpretability by applying kernel analysis techniques to this functional map. For more details on the differences, see Appendix A.2.

**Data-Similarity Kernels and Metric Learning.** The general approach of learning a data-similarity kernel is a cornerstone of statistics and machine learning, and our work is situated within this broader context (Hofmann et al., 2006; Khatib & Alkhatib, 2024). Classical methods like Principal Component Analysis (PCA) can be viewed as defining similarity through the data's covariance matrix. This was later generalized by Kernel PCA, which uses the *kernel trick* to learn non-linear similarities in a high-dimensional feature space (Schölkopf et al., 1997). A related field, metric learning, is explicitly focused on learning distance or similarity functions that are optimized for specific tasks, often by training models that pull similar data points together while pushing dissimilar ones apart (Kulis, 2013). In modern deep learning, this principle is prominent in representation learning, where models learn to project data into a latent embedding space where simple distance metrics (e.g., cosine similarity) correspond to semantic similarity (Bengio et al., 2014; Mikolov et al., 2013; Chen et al., 2020).

**Representation-Based Interpretability.** Representation-based kernels are not limited to models explicitly trained for their representations. For example, similarity measures like Centered Kernel Alignment (CKA; Kornblith et al. 2019) make it possible to derive kernels from intermediate activations of LLMs trained on next-token prediction. This falls under the broader field of representation-based interpretability, which includes other techniques such as supervised "probes" that test for specific properties of activations, and unsupervised methods, like activation atlases (Carter et al., 2019) or sparse autoencoders (SAEs; Bricken et al. 2023). As described in Section 1, these representation-based interpretability techniques offer other ways to construct kernels.

The loss kernel offers a perspective complementary to these representation-based methods. Where representation-based methods learn similarity based on what data points look like in an embedding space, the loss kernel defines similarity based on how the model treats them across the set of low-loss points. Understanding the relationship between activation-space similarity and weight-space functional coupling is a key open question. An interesting direction for future work is to bridge between these different kernel approaches. For example, Multiple Kernel Learning (MKL; Gönen & Alpaydin 2011) techniques could be adapted to learn a meta-kernel that combines information from both representations and weight-space geometry.

**Mechanistic and Causal Interventions.** Mechanistic interpretability aims to identify circuits and algorithms via targeted interventions such as activation patching and ablations (Wang et al., 2022). Our SGLD-based probe can be viewed as a complementary, weight-space analogue to these activation-space ablations. That said, our aims are different: we seek to use the loss kernel as an exploratory tool for discovering structure in data, rather than as a confirmatory tool for testing a mechanistic hypothesis.

**Developmental Interpretability.** Developmental interpretability is an approach to interpretability that models the SGD learning process as an idealized Bayesian learning process, then applies SLT to derive theoretical predictions, and finally verifies those predictions empirically on models trained using standard stochastic optimization techniques. This approach has been used successfully to detect and interpret phase transitions in stagewise learning in toy models of superposition (Chen et al., 2023; Elhage et al., 2022), transformers trained on algorithmic tasks like list-sorting and in-context regression (Carroll et al., 2025; Urdshals & Urdshals, 2025), and small language models (Hoogland et al., 2024; Wang et al., 2025b; Baker et al., 2025; Wang et al., 2025a).

The loss kernel is part of this broader agenda, particularly through its connection to key SLT quantities like the singular fluctuation (Appendix A.1).

## 6 DISCUSSION & CONCLUSION

We introduced a new technique, the loss kernel, for mapping and interpreting learned functional relationships between samples in a trained neural network. The kernel is defined as the covariance matrix of per-sample losses, computed under a distribution of parameter perturbations localized to the set of low-loss points. We first validated this method on a synthetic multitask problem, demonstrating that the kernel separates inputs by their underlying task, consistent with theoretical predictions for functionally independent mechanisms. Applied to an Inception-v1 model trained on ImageNet, we show that the loss kernel can be used to visualize the structure of the data distribution and that

this structure reflects the WordNet semantic hierarchy. These findings highlight the loss kernel as a useful practical tool for interpretability.

**Limitations.** The SGLD sampling procedure can be computationally intensive, although it is a one-time, post-hoc cost (for instance, the kernel used in the ImageNet results, Section 4, took three hours to compute on four A100 GPUs). Moreover, the results depend on the hyperparameters of the local posterior, particularly the localization strength $\gamma$ (see Appendix D.3). We also emphasize that our method is intentionally *local*, designed to interpret the specific solution found by training, not the entire global loss landscape. Finally, the kernel reveals functional correlation, not causation; it is a tool for discovering related behaviors and generating hypotheses for more targeted mechanistic investigations.

**Future Directions.** This work opens several promising avenues for future research. A primary theoretical direction is to deepen the connections to singular learning theory, and to extend this methodology beyond pairwise statistics to explore higher-order correlations. We might also hope to formalize the relationship between weight-space coupling, as measured by our kernel, and representation similarity in activation space. On an applied front, the kernel can serve as a discovery tool to guide mechanistic interpretability by identifying functionally-coupled inputs that warrant circuit-level analysis. Its ability to identify functional outliers suggests applications in anomaly and out-of-distribution detection, and the core method can be adapted to other domains like language models using token-level losses. Finally, a key direction is to apply the kernel across training checkpoints to create a *developmental* view of how a model's internal functional geometry emerges and solidifies over time.

In summary, the loss kernel offers a window into the way neural networks perceive their input data, helping to understand what data the model treats similarly, and what data the model treats differently.

## REPRODUCIBILITY STATEMENT

To ensure our work is reproducible, we provide detailed descriptions of our methodology throughout the paper and its appendices. The core SGLD-based estimation procedure for the loss kernel is formally presented in Section 2.3 and Appendix B. All experiments were conducted on a public dataset (ImageNet; Deng et al. 2009) and a standard model architecture (Inception-v1; Szegedy et al. 2014), or on a synthetic, fully described multitask arithmetic problem (Section 3 and Appendix C). A complete summary of the SGLD hyperparameters used for each experiment is available in Table 1 in Appendix B, with further implementation details and sensitivity analyses for the ImageNet setting discussed in Appendix D.3. The setup for our main ImageNet analysis, including the quantitative evaluation against the WordNet hierarchy, is detailed in Appendix D.

## LLM USAGE STATEMENT

We used Large Language Models (LLMs) to help produce this paper. We used them to edit our writing by fixing errors and improving phrasing. We also used them to brainstorm the paper's structure and get feedback on our arguments. For our experiments, LLMs helped us write code and create figures. They also assisted us in strengthening the math and proofs. The authors checked all AI-generated suggestions and are fully responsible for the content of this paper.

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

# Appendix

## A  THEORY EXTRA

### A.1  SINGULAR LEARNING THEORY

Singular learning theory (SLT) is concerned with the theory of machine learning models which are *singular*: very roughly, models for which their parameterization map is not one-to-one. Neural networks of virtually any architecture are examples of singular models. Singular models break many of the assumptions of traditional statistical learning theory (Watanabe, 2009; 2018). From an interpretability perspective, they have rich geometrical structure (e.g. in their loss landscape), which often reflects information about their internal structure (Murfet & Troiani, 2025) and their training data (Pepin Lehalleur et al., 2025).

#### A.1.1  SETUP

Classically, the setting of singular learning theory is *parametric Bayesian learning*. We review the setup here briefly. See Watanabe (2009; 2018) for a more in-depth treatment.

We begin with a parameter space $W \subset \mathbb{R}^d$ (assumed compact) and a sample space $\mathcal{X}$. A *parametric statistical model* assigns a probability $p(x|w)$ to samples $x \in \mathcal{X}$ for a given parameter $w \in W$. In singular learning theory, we typically assume that $p(x|w)$ is analytic or at the very least piecewise-analytic, which holds for most statistical models including the vast majority of neural networks.

To quantify sensitivity of $p(x|w)$ to infinitesimal parameter perturbations, we define the Fisher information matrix:

$$I_{jk}(w) = \int \left(\frac{\partial}{\partial w_j} \log p(x|w)\right) \left(\frac{\partial}{\partial w_k} \log p(x|w)\right) p(x|w)\, dx.$$

A model is **regular** at a parameter $w \in W$ if the Fisher information matrix is positive-definite at $w$, and **singular** at $w$ otherwise. We often say that a model $p(x|w)$ (without specifying any parameter $w$) is regular if it is regular for all $w$, and singular otherwise.

Note that the notion of a singular model is a purely geometric property: we have yet to discuss *learning* or *Bayesian learning*. We proceed to discuss that now. We aim to learn a data distribution

$q(x)$ over $\mathcal{X}$, which we have access to only indirectly via $n$ IID samples $\mathcal{D} = \{x_1, \ldots, x_n\}$ from $q(x)$. Our performance on this task is quantified by the *negative log-likelihood* or *training loss*, $L_n(w) = -\sum_{i=0}^{n} \log(p(x_i|w))$.

In a Bayesian setting, we have a prior distribution $\varphi(w)$, and a (tempered) posterior distribution $p(x|\mathcal{D}_n)$ obtained via Bayes rule:

$$p(x|\mathcal{D}_n) = \frac{1}{Z} \int_W \exp(-\beta L_n(w)) \, \varphi(w) \, dw$$

where $Z$ is a normalizing constant and $\beta$ is a hyperparameter known as the *inverse temperature*. When $\beta = 1$ this is the ordinary Bayesian posterior. Note that one sometimes chooses $\varphi(w)$ to be supported only in a neighborhood of a chosen point $w^* \in W$, in which case we call this a *local posterior distribution* (Lau et al., 2025).

### A.1.2 EMPIRICAL VARIANCE AND THE SINGULAR FLUCTUATION

Define the *Bayesian training error* as the *empirical* Kullback-Leibler divergence from the posterior predictive distribution to the true distribution:

$$B_t = \frac{1}{n} \sum_{i=0}^{n} \log \left( \frac{q(x)}{\mathbb{E}_w[p(x_i|w)]} \right)$$

Define the *Bayesian generalization error* as the *population* Kullback-Leibler divergence from the posterior predictive distribution to the true distribution:

$$B_t = \mathbb{E}_x \log \left( \frac{q(x)}{\mathbb{E}_w[p(x|w)]} \right)$$

The expected asymptotic difference between these quantities is given by the **singular fluctuation**:

$$\nu(\beta) = \frac{1}{2} \lim_{n \to \infty} (E_\mathcal{D}[B_g] - E_\mathcal{D}[B_t]).$$

The singular fluctuation is a birational invariant appearing in many generalization formulas within SLT, including the difference between the Bayes and Gibbs generalization errors, or the difference between the Gibbs training error and Bayes generalization error.

### A.1.3 CONNECTION TO THE LOSS KERNEL

The loss kernel can be seen as a generalization of the **empirical variance**, the empirical estimator of the singular fluctuation. The empirical variance is defined as:

$$V = \sum_{i=0}^{n} \text{Var}_w[\log(p(x_i|w)],$$

which estimates the singular fluctuation via

$$\frac{2\nu(\beta)}{\beta} = \lim_{n \to \infty} \mathbb{E}_\mathcal{D}[V].$$

If we treat the negative log-likelihood as a per-sample loss, $\ell(x; w) = -\log(p(x|w))$, and recall that the probe distribution coincides with the Bayesian posterior, this can be seen as the trace of the loss kernel evaluated on the training dataset $\mathcal{D}$:

$$V = \sum_{i=0}^{n} \text{Var}_w[\log(p(x_i|w)]$$

$$= \sum_{i=0}^{n} \text{Cov}_w[\ell(x_i; w), \ell(x_i; w)]$$

$$= \sum_{i=0}^{n} K(x_i, x_i)$$

From this perspective, the loss kernel can be seen as a *per-sample* generalization of the empirical variance, which further allows taking the covariance of two *different* samples, including possibly samples outside the training dataset $\mathcal{D}$.

### A.1.4 Towards a Population Loss Kernel

The loss kernel introduced in the main text is an *empirical* object, computed from a finite training dataset $\mathcal{D}$ of size $n$. This section sketches the link between the empirical tool and what a *population* version might look like in the limit as $n \to \infty$, which is the natural setting of singular learning theory. We expect this to be an interesting direction for future theoretical work.

**From Empirical to Population Loss.** The loss kernel probes the geometry of the *empirical loss* landscape, $L_n(w) = \sum_{i=1}^{n} \ell(x_i; w)$. In the asymptotic limit, the law of large numbers implies that this converges to a function known as the *population loss*, $L(w)$. If the per-sample loss is the negative log-likelihood $\ell(x; w) = -\log p(x|w)$, it converges to the cross entropy (equivalently, KL divergence, up to a constant) from the true distribution $q(x)$ to the model's distribution $p(x|w)$:

$$L(w) = -\int q(x) \log p(x|w) \, dx.$$

Let $L_0 = \min L(w)$. The geometry of the set $\tilde{W}_\epsilon = \{w \mid L(w) - L_0 \leq \epsilon\}$ as $\epsilon \to 0$ is intimately connected to the singularity theory of the function $L(w)$. The geometry in $L(w)$ and $\tilde{W}_\epsilon$ is rich, often reflecting interpretable computational structure, which we might hope to use for interpretability (Murfet & Troiani, 2025).

**Posterior Concentration.** The set $\tilde{W}_\epsilon$ has statistical meaning as well as geometric meaning. As the sample size $n$ goes to infinity, the posterior concentrates around $\tilde{W}_\epsilon$ for increasingly small $\epsilon$. The intuition behind this is simple (the posterior increasingly concentrates around better and better hypotheses as it gets more data), and we describe part of this connection in Appendix A.3. However, we note that actually proving convergence is highly nontrivial for singular models and that Watanabe (2009) spends multiple chapters proving similar results. From the perspective of Bayesian statistics, this convergence means that the asymptotic geometry of $\tilde{W}_\epsilon$ controls statistical quantities like the generalization error (Watanabe, 2009). For our purposes, it means that we can use the (local) posterior (the probe distribution, as we call it in the main text), whose properties can be estimated empirically using SGLD, to study the asymptotic properties of $\tilde{W}_\epsilon$.

**From Empirical Observables to Population Geometry.** We have said that one can use the local posterior (empirical) to probe the local asymptotic properties of $\tilde{W}_\epsilon$ (theoretical). To ground our discussion, we give a concrete example of how one does so for a *different* tool, the *local learning coefficient* (LLC; Lau et al. 2025). Let $B(w^*)$ be a closed ball about $w^*$. The local learning coefficient $\lambda(w^*)$ can be defined as the unique $\lambda(w^*)$ such that asymptotically as $\epsilon \to 0$:

$$\text{Vol}(\tilde{W}_\epsilon \cap B(w^*)) \approx c \, \epsilon^{\lambda(w^*)} (-\log \epsilon)^{m-1}$$

for some constant $c$ and positive integer $m$. This is the *population* quantity. It may be *estimated* in practice with a local posterior expectation value:

$$\hat{\lambda}(w^*) = n\beta \big[ \mathbb{E}_{w \sim p(w|\mathcal{D})}[L_n(w)] - L_n(w^*) \big].$$

This type of relationship is precisely what we conjecture to hold for some suitably-defined "population" version of the loss kernel.

**A Population Loss Kernel.** In this paper, we do not define a population version of the loss kernel, but we expect this to be the start of a promising direction for future work. It seems conceivable that one could define such an object, and prove that it converges to the empirical loss kernel under some limit. From this perspective, the loss kernel as we have defined it in the main text would merely be an *empirical estimator* of the population loss kernel. By analogy to quantities like the LLC, we might expect the population version to have desirable theoretical properties, such as reparameterization invariance (see Appendix C of Lau et al. 2025). Most speculatively, one might even hope that such a population loss kernel could connect to information like "computational structure" reflected in the population geometry (Murfet & Troiani, 2025).

### A.2 Training Data Attribution

The loss kernel is a natural generalization of a class of techniques known as influence functions, which are used for training data attribution (TDA; Cheng et al. 2025). This section clarifies the relationship between these objects.

### A.2.1 CLASSICAL INFLUENCE FUNCTIONS

Classical influence functions (IFs) measure how a model's parameters and, consequently, any observable quantity, would change if a single training point were infinitesimally up-weighted (Cook, 1977; Cook & Weisberg, 1982). To formalize this, consider a training set $\{z_i\}_{i=1}^n$ and a tempered empirical average loss $L_{n,\boldsymbol{\beta}}(w) = \sum_{i=1}^n \beta_i \ell_i(w)$. Let $w^*(\boldsymbol{\beta})$ be the parameter vector that minimizes this average loss. The influence of a training point $z_i$ on an observable $\phi(w)$ (e.g., the loss on a test point) is defined as the sensitivity of the observable evaluated at this new minimum to a change in the weight $\beta_i$:

$$\mathrm{IF}(z_i, \phi) := \left. \frac{\partial \phi(w^*(\boldsymbol{\beta}))}{\partial \beta_i} \right|_{\boldsymbol{\beta}=\mathbf{1}}. \tag{3}$$

Applying the chain rule and the implicit function theorem, one arrives at the well-known formula involving the Hessian of the training loss, $\boldsymbol{H}_{w^*}$:

$$\mathrm{IF}(z_i, \phi) = -\nabla_w \phi(w^*)^\top \boldsymbol{H}_{w^*}^{-1} \nabla_w \ell_i(w^*). \tag{4}$$

This approach faces significant challenges with modern neural networks, where the Hessian is typically singular (non-invertible) and computationally intractable to compute.

### A.2.2 BAYESIAN INFLUENCE FUNCTIONS

The Bayesian Influence Function (BIF) offers a principled, Hessian-free alternative (Giordano et al., 2017; Iba, 2025). Instead of tracking a single point estimate $w^*(\boldsymbol{\beta})$, the BIF measures the sensitivity of the *expectation* of an observable under a tempered posterior distribution $p(w \mid \mathcal{D}(\boldsymbol{\beta})) \propto \exp(-L_{n,\boldsymbol{\beta}}(w))\varphi(w)$:

$$\mathrm{BIF}(z_i, \phi) := \left. \frac{\partial \mathbb{E}_{w \sim p(w \mid \mathcal{D}(\boldsymbol{\beta}))}[\phi(w)]}{\partial \beta_i} \right|_{\boldsymbol{\beta}=\mathbf{1}}. \tag{5}$$

A standard result from statistical physics shows that this derivative is equal to the negative covariance over the untempered ($\boldsymbol{\beta} = \mathbf{1}$) posterior:

$$\mathrm{BIF}(z_i, \phi) = -\mathrm{Cov}_{w \sim p(w \mid \mathcal{D})}(\phi(w), \ell_i(w)). \tag{6}$$

As proposed in Kreer et al. (2025), this method can be adapted to analyze standard, non-Bayesian models by defining a *local* posterior that is constrained to the neighborhood of the trained parameters $w^*$ when combined with scalable SGMCMC-based estimators. This "local BIF" provides a practical tool for TDA that is well-defined even for singular models.

### A.2.3 CONNECTION TO THE LOSS KERNEL

The loss kernel differs from the BIF in three primary ways:

First, the BIF is *unidirectional*, measuring the influence of training points on (held-out) query points. This is because TDA focuses on *provenance*—tracing a behavior back to individual training samples. The loss kernel, in contrast, drops this distinction and directionality; it is the full symmetric, positive semidefinite kernel where entries $K(x, x') = \mathrm{Cov}[\ell(x; w), \ell(x'; w)]$ measure functional coupling between *arbitrary* inputs — whether the model has encountered those samples during training or not.

Second, while influence functions focus on *individual* interactions between (groups of) samples, the loss kernel, as a kernel, shifts the focus to *global* functional organization. By applying techniques from kernel methods (e.g., UMAP), we use the loss kernel as a primary tool for interpreting the global structure of the data manifold "as seen by the model." This comes with a caveat: it is possible to promote classical influence functions to a symmetric kernel and thereby to pull in these same kernel-derived methods. But in the classical paradigm, this operation lacks the same justification as we're able to provide for the loss kernel in Section 2 and Appendix A.1.

Finally, the loss kernel has deep theoretical grounding in singular learning theory (SLT) (see Appendix A.1). The diagonal of the loss kernel, $K(x, x)$, represents the per-sample loss variance, and its trace over the training set is an empirical variance, which is an estimator of the *singular fluctuation*, a key quantity that governs the model's generalization error. We describe this connection in more detail in Appendix A.1.3

### A.3 FROM SUBLEVEL SETS TO GIBBS DISTRIBUTION

This appendix establishes the formal relationship between expectations under the Gibbs distribution and integrals over the low-loss sets of an analytic loss function $L(w)$. We demonstrate that these quantities are related by the Laplace transform, which justifies our use of a statistical expectation about the probe distribution as a tractable tool for probing the geometry of the loss landscape.

We consider two related quantities for analyzing an observable $f(w)$. The first is the integral of $f(w)$ over the $\epsilon$-low-loss set $W_\epsilon = \{w \in \mathbb{R}^d \mid L(w) - \min_{w'} L(w') < \epsilon\}$, which defines a function of $\epsilon$:

$$g(\epsilon) = \int_{W_\epsilon} f(w) \, dw. \tag{7}$$

The second is the expectation of $f(w)$ under the Gibbs distribution $p_{\text{gibbs}}(w) = \frac{1}{Z(\beta)} \exp(-\beta L(w))$, which defines a function of the inverse temperature $\beta$:

$$\mathbb{E}_\beta[f(w)] = \frac{1}{Z(\beta)} \int_W f(w) e^{-\beta L(w)} \, dw \tag{8}$$

where $Z(\beta)$ is a normalizing constant and $W$ is the parameter space.

The following proposition details the precise relationship between $g(\epsilon)$ and $\mathbb{E}_\beta[f(w)]$.

**Proposition 1.** *The Gibbs expectation $\mathbb{E}_\beta[f(w)]$ is the Laplace transform of the low-loss integral $g(\epsilon)$, up to a known factor:*

$$\mathbb{E}_\beta[f(w)] = \frac{\beta}{Z(\beta)} \mathcal{L}\{g(\epsilon)\}(\beta), \tag{9}$$

*where $\mathcal{L}\{\cdot\}(\beta)$ denotes the Laplace transform with respect to $\epsilon$.*

*Proof.* By definition, the Gibbs expectation is given by

$$\mathbb{E}_\beta[f(w)] = \frac{1}{Z(\beta)} \int_W f(w) e^{-\beta L(w)} \, dw.$$

Using the coarea formula, we may rewrite the integral over $\mathbb{R}^d$ as an iterated integral over the level sets of the loss function:

$$\mathbb{E}_\beta[f(w)] = \frac{1}{Z(\beta)} \int_0^\infty e^{-\beta \epsilon} \left( \frac{d}{d\epsilon} \int_{L(w)<\epsilon} f(w) \, dw \right) d\epsilon.$$

Recognizing that $\int_{L(w)<\epsilon} f(w) \, dw = g(\epsilon)$, the expression becomes the Laplace transform of the derivative of $g(\epsilon)$:

$$\mathbb{E}_\beta[f(w)] = \frac{1}{Z(\beta)} \int_0^\infty e^{-\beta \epsilon} g'(\epsilon) \, d\epsilon = \frac{1}{Z(\beta)} \mathcal{L}\{g'(\epsilon)\}(\beta).$$

The derivative property of the Laplace transform states that $\mathcal{L}\{g'(\epsilon)\}(\beta) = \beta \mathcal{L}\{g(\epsilon)\}(\beta) - g(0)$. This yields:

$$\mathbb{E}_\beta[f(w)] = \frac{1}{Z(\beta)} \left( \beta \mathcal{L}\{g(\epsilon)\}(\beta) - g(0) \right).$$

The term $g(0) = \int_{W_0} f(w) \, dw$ is an integral over the set of global minima. If $L(w)$ is analytic, $W_0$ has Lebesgue measure zero, which implies $g(0) = 0$. The proposition follows. $\square$

Proposition 1 provides the theoretical basis for our methodology. The invertibility of the Laplace transform implies that the family of Gibbs expectations contains the same information as the family of low-loss-set integrals. We opt for the statistical quantity for practical reasons: $\mathbb{E}_\beta[f(w)]$ is amenable to gradient-based MCMC methods, making it computationally tractable for high-dimensional models. Furthermore, it provides a summary of the observable's behavior over all loss levels, weighted naturally by the Gibbs factor, thereby obviating the need to select an arbitrary threshold $\epsilon$. The Gibbs expectation is thus a practical and well-founded object for analyzing the properties of the low-loss subset.

### A.4 Decoupling of Disjoint Mechanisms

This section provides justification for the prediction in Section 3 that a model that has learned disjoint mechanisms for independent tasks should have zero loss covariance between samples from different tasks, under the condition that the mechanisms involve non-overlapping weights.

**Proposition 2.** *Let a model's parameters $w$ be partitioned into two disjoint sets, $w = (w_A, w_B)$. Let the training data $\mathcal{D}$ be partitioned into two disjoint sets $\mathcal{D}_A$ and $\mathcal{D}_B$, corresponding to two independent subtasks. Assume the model has learned disjoint mechanisms, such that for any data point $x \in \mathcal{D}_A$, its loss $\ell(x; w)$ is a function only of $w_A$, and for any $x' \in \mathcal{D}_B$, its loss $\ell(x'; w)$ is a function only of $w_B$. Then, under the probe distribution, the loss covariance between $x$ and $x'$ is zero:*

$$K(x, x') = \mathrm{Cov}_{w \sim p(w|\mathcal{D})}[\ell(x; w_A), \ell(x'; w_B)] = 0$$

*Proof.* Under the stated assumptions, the total loss $L(w)$ on the dataset $\mathcal{D} = \mathcal{D}_A \cup \mathcal{D}_B$ is additively separable:

$$L(w) = \sum_{x \in \mathcal{D}} \ell(x; w) = \sum_{x \in \mathcal{D}_A} \ell(x; w_A) + \sum_{x \in \mathcal{D}_B} \ell(x; w_B) = L_A(w_A) + L_B(w_B)$$

The probe distribution $p(w|\mathcal{D})$ is given by:

$$p(w|\mathcal{D}) \propto \exp(-\beta L(w)) \cdot \mathcal{N}(w|w^*, \gamma^{-1}I)$$

The spherical Gaussian localization term $\mathcal{N}(w|w^*, \gamma^{-1}I)$ also factorizes over the disjoint parameter sets:

$$\mathcal{N}(w|w^*, \gamma^{-1}I) \propto \exp\left(-\frac{\gamma}{2}\|w - w^*\|^2\right) = \exp\left(-\frac{\gamma}{2}\|w_A - w_A^*\|^2\right)\exp\left(-\frac{\gamma}{2}\|w_B - w_B^*\|^2\right)$$

Substituting the separable loss and the factorized Gaussian into the probe distribution definition, we find that the probe distribution itself factorizes:

$$
\begin{aligned}
p(w_A, w_B|\mathcal{D}) &\propto \exp(-\beta[L_A(w_A) + L_B(w_B)]) \cdot \mathcal{N}(w_A|w_A^*, \gamma^{-1}I_A) \cdot \mathcal{N}(w_B|w_B^*, \gamma^{-1}I_B) \\
&\propto \left[\exp(-\beta L_A(w_A))\mathcal{N}(w_A|w_A^*, \gamma^{-1}I_A)\right] \cdot \left[\exp(-\beta L_B(w_B))\mathcal{N}(w_B|w_B^*, \gamma^{-1}I_B)\right] \\
&\propto p_A(w_A|\mathcal{D}_A) \cdot p_B(w_B|\mathcal{D}_B)
\end{aligned}
$$

where $p_A$ and $p_B$ are the probe distributions for each sub-problem. This factorization implies that $w_A$ and $w_B$ are independent random variables under the joint posterior $p(w|\mathcal{D})$.

The covariance between the losses $\ell(x; w_A)$ and $\ell(x'; w_B)$ is defined as:

$$\mathrm{Cov}[\ell(x; w_A), \ell(x'; w_B)] = \mathbb{E}[\ell(x; w_A)\ell(x'; w_B)] - \mathbb{E}[\ell(x; w_A)]\mathbb{E}[\ell(x'; w_B)]$$

Since $\ell(x; -)$ is a function only of $w_A$, and $\ell(x'; -)$ is a function only of $w_B$, and $w_A$ and $w_B$ are independent, the expectation of their product is the product of their expectations:

$$\mathbb{E}[\ell(x; w_A)\ell(x'; w_B)] = \mathbb{E}[\ell(x; w_A)]\mathbb{E}[\ell(x'; w_B)]$$

Therefore, the covariance is zero:

$$\mathrm{Cov}[\ell(x; w_A), \ell(x'; w_B)] = \mathbb{E}[\ell(x; w_A)]\mathbb{E}[\ell(x'; w_B)] - \mathbb{E}[\ell(x; w_A)]\mathbb{E}[\ell(x'; w_B)] = 0$$

This holds for any $x \in \mathcal{D}_A$ and $x' \in \mathcal{D}_B$. $\qquad\square$

While this sketch is illustrative, we note that it may be somewhat unrealistic to believe that deep learning models implement distinct mechanisms in disjoint sets of weights. See for instance the phenomenon of *polysemanticity* (Elhage et al., 2022). It may require a change of coordinates before mechanisms cleanly factorize. From a singular learning theory perspective, the correct remedy here is likely found at the level of *population* quantities, which are often invariant to arbitrary (diffeomorphic) coordinate change (see for example Appendix C of Lau et al. (2025)). We discuss the possibility of a *population* loss kernel with such a property in Appendix A.1.4, but we largely leave that to future work.

# B  STOCHASTIC-GRADIENT MCMC ESTIMATOR

Evaluating the *loss kernel* $K(x, x') = \text{Cov}_w[\ell(x; w), \ell(x'; w)]$ requires Monte-Carlo samples from the probe distribution $p(w \mid \mathcal{D})$. Following Lau et al. (2025), we use Stochastic Gradient Langevin Dynamics (SGLD; Welling & Teh 2011).

**Update rule.**  With stochastic mini-batch $B_t \subset [n]$ of size $m$ and step size $\epsilon$, SGLD performs

$$w_{t+1} = w_t - \frac{\epsilon}{2}\Big(\frac{n}{m}\sum_{x \in B_t}\nabla_w\ell(x; w_t) + \gamma\big(w_t - w^*\big)\Big) + \sqrt{\epsilon}\,\xi_t, \qquad \xi_t \sim \mathcal{N}(0, I). \tag{10}$$

The first term is the stochastic gradient of the loss; the second is the gradient of the Gaussian localization potential $\frac{\gamma}{2}\|w - w^*\|^2$; the injected Gaussian noise ensures asymptotic convergence to $p(w|\mathcal{D})$ as $\epsilon \to 0$.

**Parallel chains and burn-in.**  To improve mixing we run $C$ independent chains, each initialized at $w^*$. After discarding a burn-in of $b$ iterations, we retain $T$ draws $\{w_{c,t}\}_{t=1}^{T}$ per chain. For every retained weight we record the vectors $\ell(x_i; w_{c,t})$.

**Estimators.**  The unbiased plug-in estimators for $K(x, x')$ and $R(x, x')$ are:

$$\hat{K}(x, x') = \frac{1}{CT - 1}\sum_{c=1}^{C}\sum_{t=1}^{T}\big(\ell(x; w_{c,t}) - \hat{\mu}(x)\big)\big(\ell(x'; w_{c,t}) - \hat{\mu}(x')\big),$$

$$\hat{R}(x, x') = \hat{K}(x, x')/\sqrt{\hat{K}(x, x)\hat{K}(x', x')},$$

where $\hat{\mu}(x)$ is the estimated average loss:

$$\hat{\mu}(x) = \frac{1}{CT}\sum_{c,t}\ell(x; w_{c,t}).$$

**Batched evaluation.**  At each retained iteration $w_{c,t}$, a full forward pass is performed over the entire dataset of interest to compute and store the loss vector $(\ell(x; w_{c,t}))_i$. In contrast, the SGLD update in Equation (10) only requires a single backward pass on a small, random minibatch $B_t$.

Contrast this with the local Bayesian Influence Function (BIF; Kreer et al. 2025), which requires computing forward passes over two separate "attribution" and "query" datasets. We compute forward passes only over a single set, yielding an $n \times n$ covariance kernel. This is effectively the same as treating every sample as both an "attribution" and a "query" point to measure the functional coupling between all pairs of inputs.

**Avoiding Spurious Correlations.**  We observe that a high correlation between inputs of the same label often is spurious. At some SGLD hyperparameters, noise injected in the unembedding weights causes inputs of the same label to always slightly increase or decrease in loss together. This can dominate the observed correlations. Similar issues apply to per-sample gradient and activation based methods, where often the unembedding weights aren't used in computation for the same reason. For example, we find that we can perfectly recover input labels by running SGLD for 10 steps on an *untrained model*. UMAP works via a fuzzy nearest neighbors lookup, and so to deconfound our UMAPs we delete edges between same label inputs during the neighbor finding step. This means two inputs of the same label will never be neighbors *just because they share a label*.

## HYPERPARAMETERS OVERVIEW

Table 1 summarizes the hyperparameter settings for the correlation kernel experiments. We sample with SGLD: $m$ is the batch size, $C$ is the number of chains, $T$ the number of draws per chain, $b$ is the number of burn-in steps, $\epsilon$ is the learning rate, $\beta$ is the inverse temperature, and $\gamma$ is the localization strength.

**Table 1:** Summary of hyperparameter settings for correlation kernel experiments. Hyperparameters are defined in Appendix B and Section 2.3.

| Section | Model | Dataset | $m$ | $C$ | $T$ | $b$ | $\epsilon$ | $n\beta$ | $\gamma$ |
|---|---|---|---|---|---|---|---|---|---|
| Section 3 | 2 Layer Transformer | Modular Addition and Modular Division mod 97. | 512 | 30 | 800 | 200 | $2 \times 10^{-7}$ | 500 | 30,000 |
| Section 4 | InceptionV1 | ImageNet | 256 | 15 | 500 | 100 | $5 \times 10^{-5}$ | 20 | 4,000 |
| Appendix D.4 | InceptionV1 | ImageNet with 1,000 random samples mislabeled. | 256 | 8 | 1000 | 100 | $1 \times 10^{-5}$ | 20 | 4,000 |
| Appendix D.3 | InceptionV1 | ImageNet | 256 | 5 | 1200 | 2000 | $1 \times 10^{-4}$ | 20 | Varied |

## C  SYNTHETIC TASK EXTRA

This section provides additional details for the synthetic multitask experiment presented in Section 3.

**Model Architecture.**    We use a two-layer transformer with the same architecture as that used in the original grokking experiments by Power et al. (2022). We refer the reader to their work for specific architectural details. We make one modification which is to double the vocabulary, so that each task uses an independent set of tokens.

**Tasks and Dataset.**    The model was trained on a multitask problem comprising modular addition and modular division, both over the prime modulus $p = 97$. Inputs for both tasks are sequences of the form a, b, result. The use of non-overlapping vocabularies is sufficient to for the model to distinguish which operation must be performed.

Training and evaluation data were generated by sampling integers $a, b \in \{0, \ldots, 96\}$ uniformly at random. For modular division $a/b$, we compute $a \cdot b^{-1} \pmod{97}$, where $b^{-1}$ is the modular multiplicative inverse of $b$. We exclude cases where $b = 0$.

**Training.**    The model was trained on both tasks simultaneously using the Adam optimizer until it achieved 100% accuracy on the training set.

**Loss Kernel Estimation.**    After training, we estimated the loss kernel to analyze the learned functional structure. We used SGLD to draw samples from the local posterior distribution, localized around the final trained weights $w^*$. We collected a total of 30,000 posterior weight samples after an initial burn-in period of 200 steps for each chain. The loss kernel was then computed over an evaluation set of 10,000 randomly selected inputs, evenly split between the modular addition and modular division tasks. The specific SGLD hyperparameters, including learning rate $\epsilon$, inverse temperature $\beta$, and localization strength $\gamma$, are provided in the main hyperparameter summary (Table 1).

## D  IMAGENET EXTRA

### D.1  INCEPTIONV1

We apply our method to InceptionV1 (Szegedy et al., 2014). Each InceptionV1 experiment *evaluates* posterior correlations over 10,000 ImageNet validation samples, while sampling over the full ImageNet (Deng et al., 2009) training dataset. To reduce memory overhead, we downscale all images to 256x256 resolution. Full hyperparameters are included in Table 1. We find that the quality of correlations depends significantly on total draws used: see Appendix D.3 for extended discussion.

### D.2  QUANTIFYING HIERARCHICAL STRUCTURE

To move beyond visual inspection, we quantitatively assess how well the kernel's structure aligns with the WordNet hierarchy.

**Taxonomic lift construction.**    For each validation image $i$ with WordNet label $y(i)$ at depth $d(i)$, we take its top-$k$ neighbors under the correlation kernel $R$ (we use $k{=}30$). For any candidate ancestor depth $d'$, define

$$\mathrm{Lift}(d, d') \;=\; \frac{\Pr[\exists\, a \text{ at depth } d' \text{ such that } a \preceq y(i) \;\wedge\; a \preceq y(j) \mid j \in \mathrm{NN}_k(i),\ d(i){=}d]}{\Pr[\exists\, a \text{ at depth } d' \text{ such that } a \preceq y(i) \;\wedge\; a \preceq y(j)]},$$

where $a \preceq y$ means "$a$ is an *ancestor* of label $y$" in the ImageNet–WordNet hierarchy (equivalently, $y \in \text{Descendants}(a)$). We condition on query depth $d$ to avoid confounding from the uneven leaf-depth distribution. Curves in Fig. 5 report $\text{Lift}(d, d')$ versus the depth of the shared ancestor (i.e., tree distance from the root), with one curve per query depth $d$.

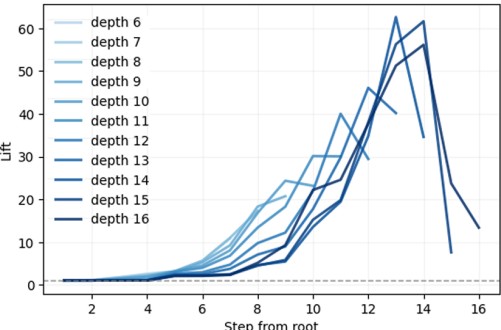

**Figure 5: Taxonomic Lift vs. Hierarchy Depth.** Lines depicts the weighted probability (lift) that the nearest neighbors of an input with a label $d$ nodes deep in the WordNet hierarchy will share a parent node at depth $d'$. The $x$–axis is the WordNet tree distance (edges) from root to the shared ancestor. We report *lift* as the ratio of this probability to the dataset base rate at depth. See Appendix D.2 for details.

**Estimation details.** We evaluate on $n{=}10{,}000$ validation images and average the probability over queries with depth $d$. We exclude identical-label pairs when constructing neighbors to avoid trivial lifts; The decrease towards the end of lines for larger $d$ is because nodes deep in the hierarchy often have few children.

**Interpretation.** Lift $> 1$ indicates that kernel-nearest neighbors are *more likely than chance* to share a taxonomy node at depth $d'$. We observe: (i) lift increases with query depth $d$ (deeper, more specific classes show stronger taxonomic cohesion); (ii) lift peaks at intermediate $d'$ and tapers near the root (ancestors too coarse) and near leaves (sparsity reduces shared-ancestor opportunities), consistent with the qualitative UMAP and the correlation–distance decay in the main text.

### D.3 Hyperparameter Dependence

**Convergence of the Estimator.** Centered Kernel Alignment (CKA) is a similarity measure between two kernel (or Gram) matrices. Given kernels $K, L \in \mathbb{R}^{n \times n}$, the CKA is defined as

$$\text{CKA}(K, L) = \frac{\langle K_c, L_c \rangle_F}{\|K_c\|_F \, \|L_c\|_F},$$

where $K_c$ and $L_c$ denote the centered versions of $K$ and $L$, and $\langle \cdot, \cdot \rangle_F$ is the Frobenius inner product. This normalization ensures that $\text{CKA}(K, L) \in [0, 1]$, with 1 indicating identical representational structure.

CKA analysis reveals the consistency and similarity of representations across different training runs and sampling procedures. We can use it to compare the kernels we get at different hyperparameters, but also how the kernel evolves as total SGLD step count increases. Figure 6 **A** shows how the CKA between the kernel at step $t$ of SGLD and the kernel at the final step changes as a function of $t$. Note that $t$ is total steps over all chains and that we limit individual chain. We find that higher $\gamma$ leads to faster convergence. Similarly, Figure 6 **B** shows the CKA between kernels computed using different $\gamma$ parameters. At high $\gamma$ the CKA between kernels is close to 1, meaning the kernel is robust to specific choice of $\gamma$.

**The effect of $\gamma$.** Recall that the hyperparameter $\gamma$ controls how tightly the probe distribution is concentrated around $w^*$ in parameter space. Empirically, Figure 6 **D** quantifies this trade-off with a simple *lift* metric (the weighted probability that a sample's nearest neighbors under the loss kernel $R$ share an attribute, divided by that attribute's base rate). At *low* $\gamma$, neighbors are disproportionately matched by low-level cues such as *color* (high color-lift); as $\gamma$ increases, color-lift falls while *hierarchical* coherence (neighbors sharing nearby nodes in WordNet) rises sharply. We detail how we group inputs by color in Appendix D.6 – we use the groupings to compute the same way we compute per-node lift in Appendix D.2.

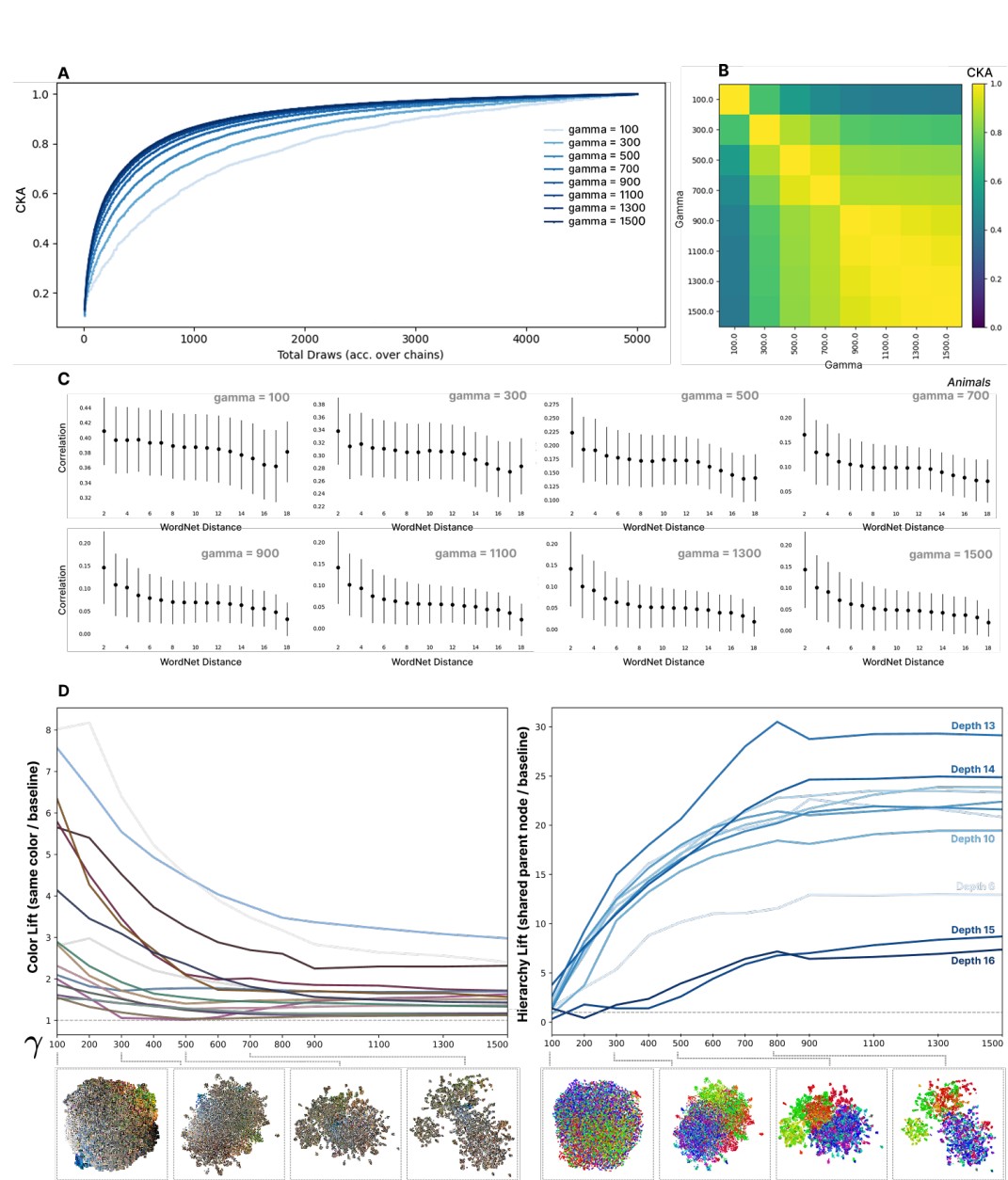

**Figure 6: Dependence of the kernel on the SGLD hyperparameter** $\gamma$**. A**: CKA between the kernel at step $t$ of SGLD and the final step of SGLD. Shows how the kernel converges as a function of total draws taken. **B**: CKA between kernels computed using different $\gamma$ values.The kernel stabilizes between $\gamma = 900$ and $\gamma = 1500$. **C**: Loss kernel correlation vs distance across the WordNet hierarchy for *animal* inputs. As $\gamma$ increases inputs closer in the hierarchy become relatively more correlated than inputs further away in the hierarchy, showing that gamma controls how reflected the hierarchy is in the kernel. **D**: Lift (neighbor match rate divided by base rate) for *color* (left) and *ImageNet-WordNet node* (right) as $\gamma$ varies. Low $\gamma$ emphasizes low-level cues (high color-lift); increasing $\gamma$ suppresses color-lift while strongly increasing hierarchical coherence. UMAPs beneath each curve illustrate the same trend qualitatively.

UMAP snapshots beneath each curve show the same transition qualitatively: low $\gamma$ yields broad, texture/color-organized neighborhoods, while high $\gamma$ foregrounds semantically tight groupings aligned with the taxonomy. Specific per-experiment hyperparameter settings are detailed in the below section.

### D.4 DETECTING MEMORIZATION

We test whether the loss kernel is sensitive to changes in the functional constraints imposed on the model by making a targeted change to the model's training data distribution. We randomly mislabel a subset of the training data, forcing the model to memorize in order to achieve a low loss.

Memorization imposes a strict *functional constraint* on our model. A very precise weight setting is required to achieve high performance – put simply, the set of parameters that achieve low loss on the mislabeled set forms a much narrower region (a sharper basin) than the region that preserves low loss when the mapping can be supported by shared features.

As detailed in Appendix A.1, the trace of our kernel is an estimator for the singular fluctuation, a quantity that appears in the asymptotic

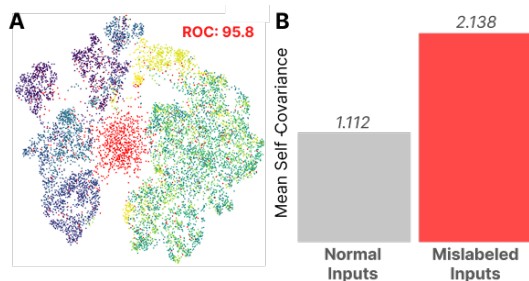

**Figure 7:** **A**: A UMAP visualization of the loss kernel for an Inception-v1 model trained until convergence on ImageNet with $1,000$ samples mislabeled. Mislabeled inputs (red) form a distinct cluster. We report an ROC of 95.8 for detecting mislabeled points using per-sample loss variances $l$. **B**: The mean self-covariance, or singular fluctuation, of normal (1.112) and mislabeled (2.138) inputs.

formula for the Gibbs generalization error. The kernel itself can be seen as measuring the first-order change in a related quantity known as the Bayes generalization error, with respect to the importance of each data point. While these notions of generalization are not immediately related to the type of memorization we study empirically, this provides some intuitive support to the idea that memorized examples will show up with a large self-correlation $K(x, x)$.

### D.5 THE LOSS KERNEL OVER DEVELOPMENT.

To visualize how functional geometry emerges during learning, we compute the loss–correlation kernel at fixed training checkpoints and embed the induced distances $d(x, x') = 1 - R(x, x')$ with the *same* UMAP hyperparameters across time (and with same–label edges removed; see Appendix B).

Figure 8 shows a coarse-to-fine trajectory. Bar a handful of curious outliers, at initialization the kernel is essentially structureless. We leave study of these outliers to future work. By step 710, a weak global anisotropy appears that roughly separates animate from inanimate classes. By step 1388, coherent clusters begin to form (e.g., *dogs*). At step 3290, multiple subgroups sharpen and separate, and by step 5298 the geometry stabilizes into well-defined, semantically coherent regions that mirror the WordNet hierarchy.

### D.6 QUANTIFYING COLOR LIFT.

We describe our method for computing the average per-color lift as shown in Figure 6 and Figure 9. In order to compute the lift we must bucket images into discrete color groups. To do so, for each input image, we compute

$$\boldsymbol{\mu}_i = \left( \frac{1}{P} \sum_{p=1}^{P} R_{i,p}, \ \frac{1}{P} \sum_{p=1}^{P} G_{i,p}, \ \frac{1}{P} \sum_{p=1}^{P} B_{i,p} \right),$$

where $P$ is the number of pixels in image $i$, and $R_{i,p}, G_{i,p}, B_{i,p}$ are the red, green, and blue values of pixel $p$ in image $i$. (Equivalently, $\boldsymbol{\mu}_i = (\overline{R}_i, \overline{G}_i, \overline{B}_i)$, where each bar denotes the mean over all pixels in image $i$.) 2) Cluster the set $\{\boldsymbol{\mu}_i\}$ into $k$ groups using *farthest point sampling* (FPS). FPS ensures that cluster centers are spread out over the uneven distribution of RGB means (e.g. many gray/brown tones).

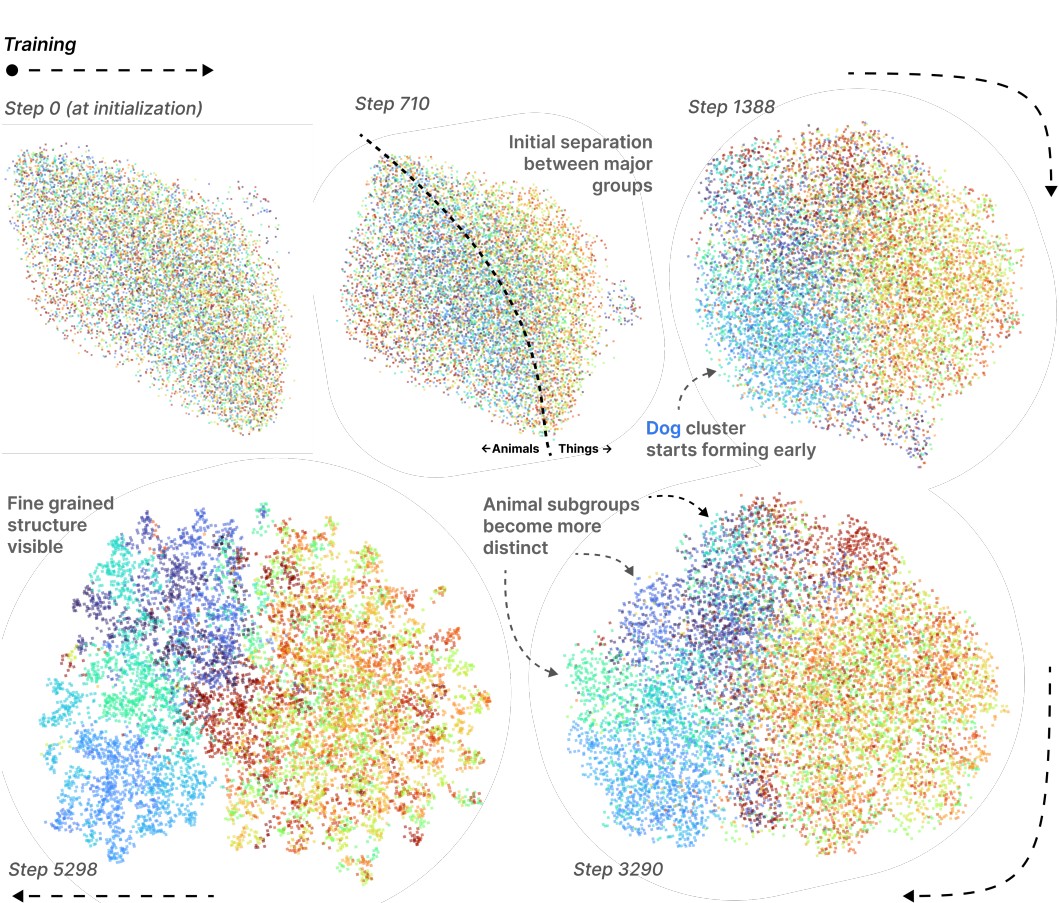

**Figure 8: Evolution of the kernel over training.** UMAPs of the loss kernel taken at various steps over training, for an InceptionV1 model trained on ImageNet. Between **initialization** (top left) and **step 710** (top middle) the model begins to distinguish between animals and things – A gradient of differentiation is established. At step 1388 (top right) significant structure is apparent, with *Dogs* forming an early cluster. **Step 3290** (bottom right) sees many subgroups forming distinct clusters. By **step 5298** (bottom left) the kernel is fully formed.

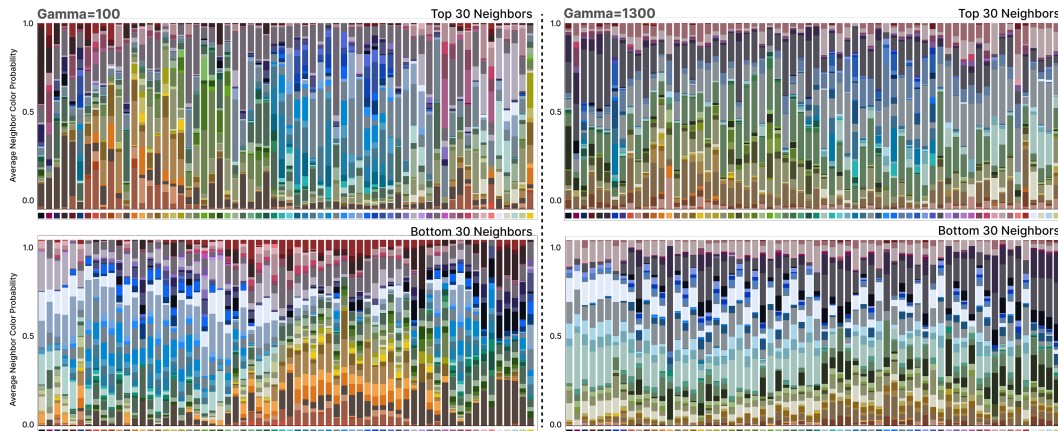

**Figure 9: Average color–neighbor probabilities, for low and high Gamma.** Stacked barchart versions of transition matrices, where a transition can be made from an input to its top (first row) or bottom (second row) 30 correlated inputs. The probability of a transitioning from an image "close to color" $A$ on the x axis to an image "close to color" $B$ is given by the height of $B$'s bar in the stack. The right column shows the transition matrix obtained when using $\gamma = 100$ during sampling, while the right shows the results for $\gamma = 1300$. For $\gamma = 100$ we see significant color striation in both rows, especially in the bottom correlated inputs (e.g. blue inputs have pronounced low correlation with orange inputs). Contrastingly patterns visible in $\gamma = 1300$ are much more uniform.

## D.7 EXTRA IMAGENET EXAMPLES

We provide more examples of the top correlated inputs from the visualization experiment in Section 4 and Figure 4. These inputs were randomly selected in chunks of 10 from between the 600th and 700th inputs of the 2500 for which we computed the loss kernel. The full set top-correlated inputs for all 2500 inputs is available at `https://github.com/singfluence-anon/sf_imagenet_corrs`.

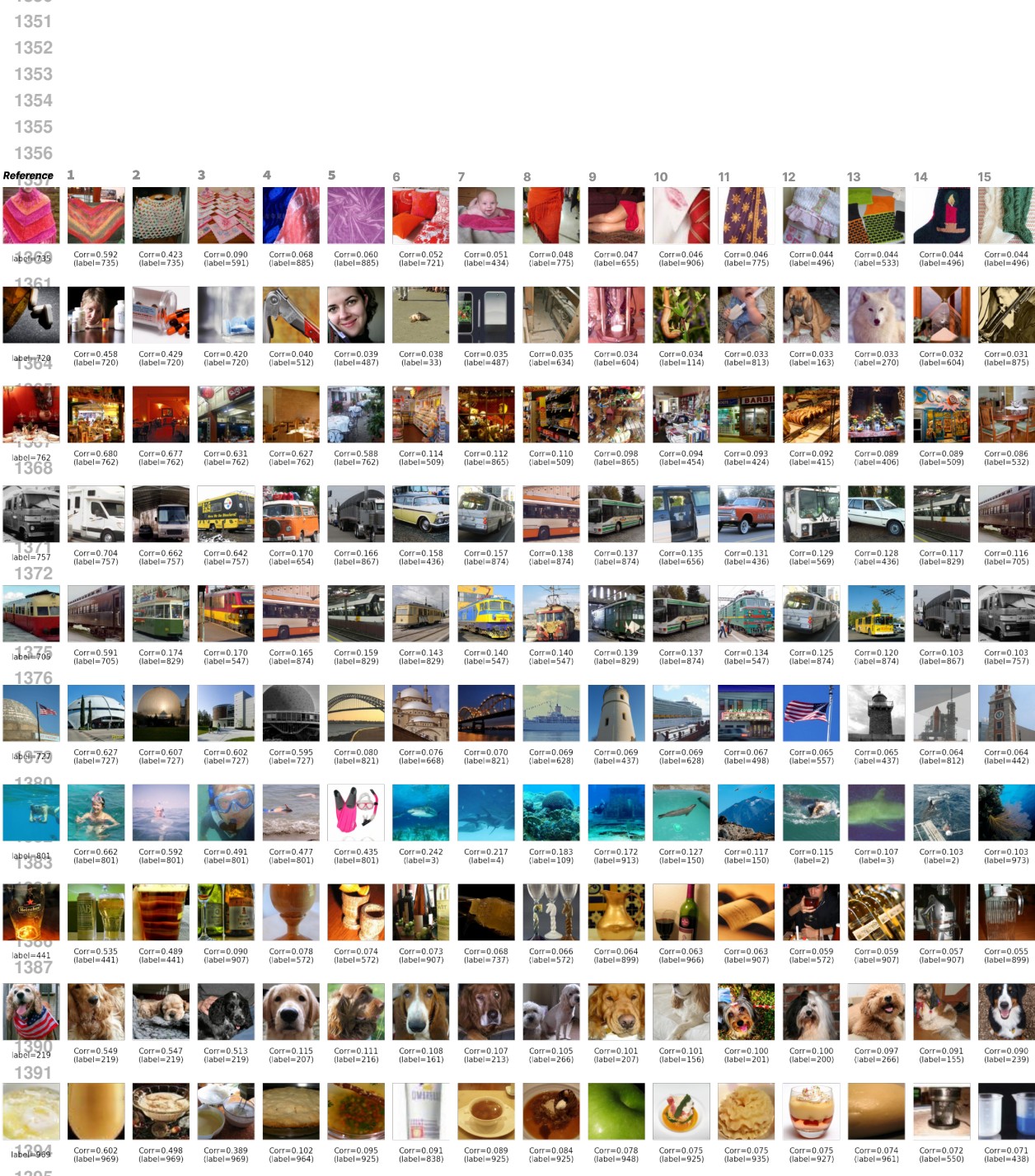

**Figure 10: Top 15 Correlated Inputs With Reference Input (randomly selected references).** Reference images are the leftmost column.

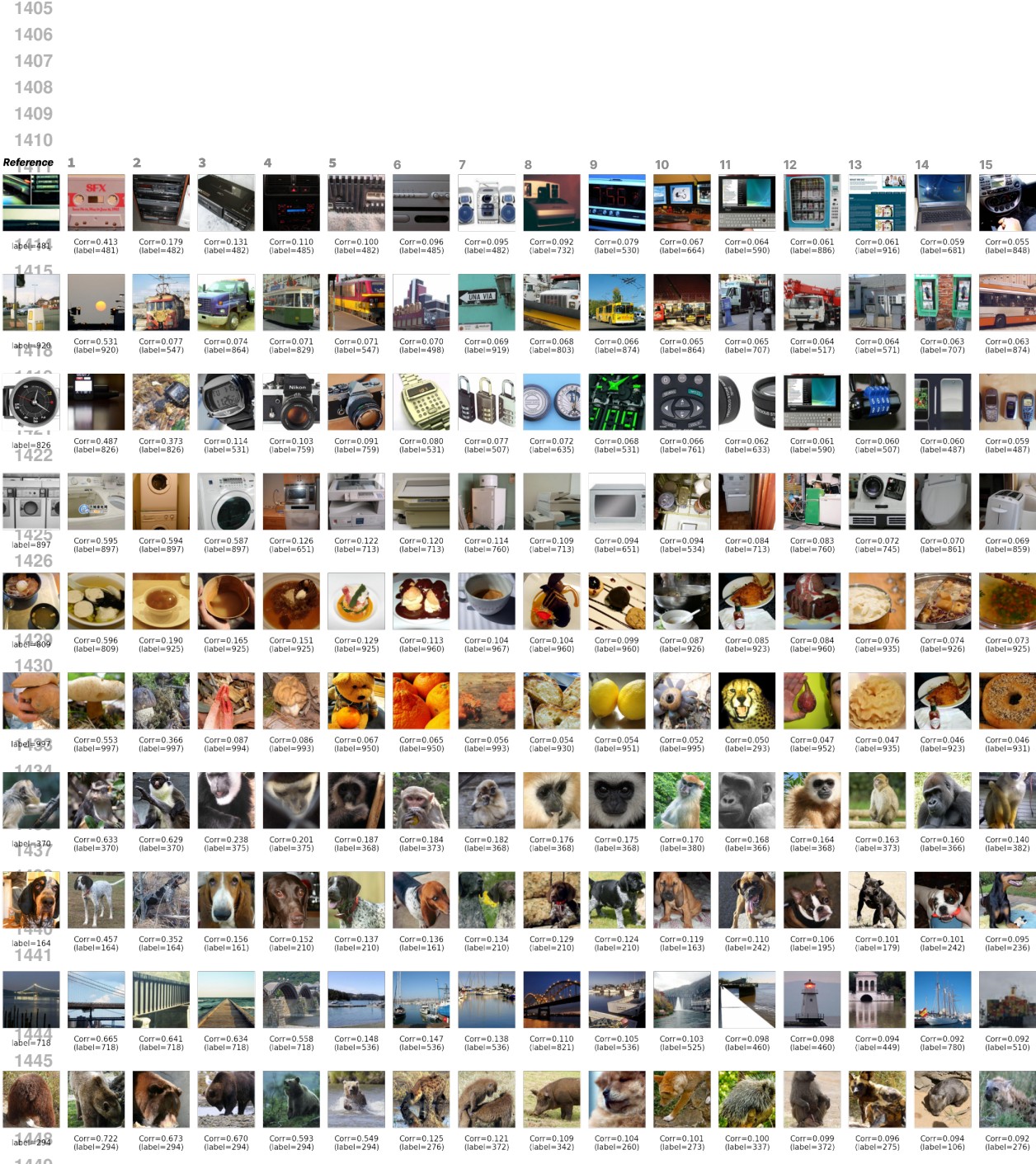

**Figure 11: Top 15 Correlated Inputs With Reference Input (randomly selected references).** Reference images are the leftmost column.

