# OpenReview forum: "The Loss Kernel: A Geometric Probe for Deep Learning Interpretability"
_ICLR.cc/2026/Conference — ICLR 2026 Conference Withdrawn Submission_

### Official Review · Reviewer_HjYs · 2025-10-29

**Soundness:** 3
**Presentation:** 2
**Contribution:** 3
**Rating:** 4
**Confidence:** 3

**Summary:**

This paper proposes an interpretability method called the loss kernel, which measures the similarity between data points based on a trained neural network. The loss kernel can serve as a measure of functional coupling and is validated under controlled experimental settings.

**Strengths:**

1. This paper propose a  interpretability method called loss kernel, which is able to measuring similarity between data instance.
2. The visualizations are presented quite well, especially those in the Appendix (pages 24–27).
3. This paper provides a clear explanation in the Limitations and Reproducibility Statement sections regarding the drawbacks introduced by SGLD and the settings for reproduction. This transparency greatly facilitates the reproducibility of the work and supports future extensions.

**Weaknesses:**

1. The paper lacks more practical real-world application cases. In practice, instance similarity has clear use cases in several important research areas, such as transfer learning, continual learning, few-shot learning, and federated learning. The paper is suggested to clarify how the proposed interpretability method can concretely guide algorithmic or experimental design. For example, does it help mitigate data drift or alleviate catastrophic forgetting? This aspect requires further discussion. I suggest that the authors select one representative scenario or task among these domains to demonstrate how the proposed loss kernel exhibits its advantages. For instance, a relevant study [1] illustrates how data similarity can be used to enhance federated learning.

2. In the third paragraph of the Introduction, the authors mention the loss landscape. It is recommended to visualize the loss landscape, as doing so would make the paper more convincing and help readers better understand the behavior of the proposed loss kernel.

3. Overall, the paper lacks sufficient empirical or conceptual persuasiveness. Although the proposed loss kernel is presented with reasonable background and motivation, its usability and effectiveness currently remain theoretical. While extensive experiments may not be necessary, the authors should clearly articulate what insights or practical guidance their work offers for neural network training or optimization.

4. This paper should explicitly explain why Stochastic Gradient Langevin Dynamics (SGLD) is used in Section 3.2 to generate the set $\{w_s\}_{s=1}^S$, even though the resulting samples appear to deviate from the target posterior distribution $p(w \mid \mathcal{D})$. The paper should clearly describe how $\{w_s\}_{s=1}^S$ is related to $p(w \mid \mathcal{D})$. Is this set directly assumed to follow the conditional distribution, or is it merely an approximation to it?

5. This paper employs SGLD to estimate $p(w \mid \mathcal{D})$.  However,  SGLD may not an especially efficient algorithm, and its accuracy may not surpass that of other gradient-based methods. How do the authors justify this choice? Since SGLD is a stochastic differential equation, have the authors verified the theoretical conditions necessary for its ergodicity?

6. In Proposition 2, the model parameters are divided into two disjoint subsets, namely $w_A$ and $w_B$. The authors should clarify why it is always possible to partition the parameters in this way, or specify under what assumptions such a separation holds.

[1] *Efficient Distribution Similarity Identification in Clustered Federated Learning via Principal Angles Between Client Data Subspaces*, AAAI 2023.

**Questions:**

Is this set $\{w_s\}_{s=1}^S$ directly assumed to follow the conditional distribution$p(w \mid \mathcal{D})$, or is it merely an approximation to it?

---

### Official Review · Reviewer_dMBY · 2025-10-31

**Soundness:** 2
**Presentation:** 2
**Contribution:** 3
**Rating:** 2
**Confidence:** 2

**Summary:**

This paper proposes the loss kernel, an interpretability method for measuring similarity between data points according to a trained neural network. The loss kernel is inspired by singular learning theory, and it measures two inputs as similar if they are processed in similar ways by a given trained neural network. To validate the effectiveness of the loss kernel, the authors conduct a synthetic multitask experiment, and show that the loss kernel can separate subtasks. Then, the loss kernel is applied to Inception-v1 on ImageNet as an interpretability tool, and shows a coherent semantic structure in the data.

**Strengths:**

- This paper proposes an innovative method to measure input similarity. Rather than using activations or representations, the proposed loss kernel originates from the loss landscape, and measures similarities through whether two inputs are processed in similar ways by a neural network.
- This paper discusses the theoretical connection between the proposed loss kernel and singular learning theory and influence functions.

**Weaknesses:**

- The organization of this paper has a problem. Many theoretical discussions are presented in the appendix, leaving insufficient content in the main body (less than 7 pages, excluding the related works and conclusion sections at the end).
- The experiments are limited to one synthetic task and Inception-v1 on ImageNet. It remains unknown if the conclusions could generalize beyond multitask arithmetic, or to other network architectures and real datasets.
- Lacking discussions on why two inputs can be regarded as semantically similar if they have similar behaviors in the loss landscape.
- In Figure 2, the positive correlation does not seem significant. How much is the $R^2$ of the linear regression?
- Sensitivity analysis of $\beta$ is lacking.

**Questions:**

- In Figure 3(D), do the three subplots share the same x-axis? Although these distributions have different shapes, they do not seem to be separable if mixed together.
- In Eq (2), why do you adopt the covariance? (not KL-divergence or other metrics?)

---

### Official Review · Reviewer_8jkJ · 2025-11-01

**Soundness:** 3
**Presentation:** 3
**Contribution:** 2
**Rating:** 4
**Confidence:** 3

**Summary:**

The paper defines the loss kernel K(x,x′) as the covariance of per-sample losses under a local, low-loss probe distribution over parameters. Practically, the probe is a tempered Gibbs posterior re-centered at the trained weights w∗, sampled via SGLD. The authors (i) motivate the construction from singular learning theory, (ii) validate on a two-task synthetic arithmetic setting where independent mechanisms should yield near-zero cross-task covariance, and (iii) map 10k ImageNet examples with Inception-v1, showing UMAP embeddings whose blocks correlate with WordNet hierarchy structure.

**Strengths:**

1. The probe distribution (tempered Gibbs × local Gaussian around w∗) is well-motivated and yields a PSD kernel by design (covariance). The normalized correlation form R is affine-invariant in the loss.

2. The paper connects the kernel’s diagonal/trace to empirical variance and the singular fluctuation, and positions the method within singular learning theory with population-limit discussion in the appendix.

3. The ImageNet maps and nearest-neighbor examples under R are interpretable and show coarse-to-fine semantic organization aligned with WordNet; training-trajectory snapshots suggest developmental structure.

**Weaknesses:**

1. Some typos in this paper:
- Figure 1 caption uses “InceptionV1” while the text uses “Inception-v1” (style inconsistency).
- Figure 1 caption spells “diaspids”; later text uses “diapsids” (misspelling).
- Extra space before comma in examples: “wire-haired fox terrier , “goldfish” …”.
2. The training loss is written as a sum from i=0 to n (appears multiple times), which implies n+1 terms; standard is i=1…n (or 0…n−1). Please fix for consistency across (2.1) and Appendix A.1.2.
3. The normalized kernel sets R(x,x′)=0 when either variance term K(x,x) or K(x′,x′) is zero; justify this convention and discuss what it means for points with exactly zero variance under the probe (e.g., deterministic losses).
4. Appendix A.3 (“From sublevel sets to Gibbs distribution”) is truncated mid-sentence in the current draft (“The second is the …”), and the main text references a formal relationship via Laplace transforms without showing a complete statement or conditions (analyticity, tails). Please complete the statement and specify assumptions.
5. The PSD claim for K is correct as a covariance, but the paper could add a one-line proof or citation and state conditions under which SGLD plug-in estimators preserve PSD up to sampling error (numerical symmetrization).
6. Conceptually, the method is a symmetric generalization of Bayesian Influence Functions to a pairwise covariance kernel plus a local posterior; the paper should deepen comparisons to activation-space kernels (CKA, cosine in intermediate layers) and to “influence-as-kernel” baselines to delineate when weight-space coupling gives strictly new insights.
7. The ImageNet study shows strong visuals, but quantitative alignment with WordNet is only mentioned; please report concrete metrics (e.g., rank correlation between kernel distances and WordNet distances, cluster purity/NMI at multiple granularities) and error bars across SGLD seeds.

**Questions:**

See Weaknesses.

---

### Official Review · Reviewer_11tb · 2025-11-02

**Soundness:** 2
**Presentation:** 3
**Contribution:** 2
**Rating:** 4
**Confidence:** 3

**Summary:**

This paper introduces the method of loss kernel to measure the similarity between examples. The method is designed for interpreting the data structure according to a trained neural network. It first assumes the parameter $w$ to follow a tempered Bayesian posterior with Gaussian prior. It considers both the low-loss constraint and locality constraint. Then the loss kernel is defined as the covariance matrix of per-sample losses under the parameter distribution. The parameters are empirically sampled by SGLD. The method is evaluated by a synthetic task and the real ImageNet task. The loss kernel extracted reasonable global structures, i.e., data points with related semantics are distributed into clusters, which fit human expectations and interpretations.

**Strengths:**

1. The paper proposed an interesting idea of interpreting the functional coupling between samples by adapting BIF.
2. The method is based on well-established theories.
3. Analyzing the data structure by measuring the pair-wise functional coupling is an unexplored area and could potentially provide new insights for DNN interpretability.
4. The paper is well written and easy to understand. Necessary background knowledge is introduced.

**Weaknesses:**

1. The technical contribution is a bit weak as it’s mostly a direct application of the BIF method. The authors explain the difference between loss kernel and BIF in A.2.3, but the claimed differences do not seem significant.
2. Figure 1 shows the geometry of the loss kernel on ImageNet. The formed clusters are shown to align well with the hierarchical semantics. However, this can also be achieved by simply using feature similarity. The paper needs to provide empirical results to compare the proposed loss kernel against feature similarity.
3. Experiments are not solid enough. Only two datasets are used, and the model Inception-v1 is too old.
4. The findings on ImageNet are not surprising. The results don’t show any new insights for interpreting the model or dataset. Is there any real-world application to showcase the usage of the loss kernel results?

**Questions:**

See Weaknesses

---

### Note · Authors · 2025-11-22

**Comment:**

We thank the reviewers for their feedback, and plan to incorporate their comments into a future version of this work. We also thank the reviewers for their appreciation of the work, including identifying the method as “well-founded” and “innovative”, and the presentation as “well-written and easy to understand.” However, we have decided at this time to withdraw the submission.

For the benefit of the public record, we wish to clarify two specific points raised by reviewers.

**Similarity to the local BIF**
Reviewers 11tb and 8jkJ noted similarities between our method and the (local) BIF. We address the distinction in Appendix A.2.3, but it is worth expanding on the distinction and our contribution here. While the mathematical machinery (posterior covariance) is shared, the purpose behind the two methods differs, leading to distinct theoretical objects and substantially different application and interpretation.

The BIF (and TDA in general) is fundamentally about *provenance* \- attributing model behavior as being *caused* by specific data points. This can be seen in the metrics used to evaluate TDA, such as leave-one-out retraining or the linear datamodeling score \- the purpose is to determine whether the presence or absence of a data point in a model’s training data counterfactually influenced the final behavior of the model. Ultimately, this is a claim about the training process, rather than purely a claim about the structure of the trained model. It is also why the BIF is only defined for points in the training set of the model.

Our goal with the loss kernel is different, and correspondingly the definition. We aim to use the loss kernel as a tool to investigate the learned representations of the final trained model (i.e. interpretability), which is why, *unlike the BIF, the loss kernel can be sensibly defined on samples outside the training set of the model*.

It is also why we may treat the resulting covariance matrix not as a collection of individual attributions but as a symmetric PSD kernel defining a similarity structure on the input space. That is, the loss kernel has a *geometric* interpretation that the BIF does not. We believe that this is a substantial contribution of our work, allowing us to apply standard kernel analysis techniques like UMAP to visualize this data similarity structure. The local BIF has no such geometric interpretation.

Finally, we emphasize the connection of the loss kernel to singular learning theory, which we discuss in Appendix A.1. This is currently nascent in the present version of the work, but we believe it offers substantial promise for theoretically grounding interpretability, and we plan to expand on this connection in future versions of the work. To explain the motivation, at the risk of excessive speculation: existing work connects (in a theoretical setting, to a limited extent) singularity structure of the loss function to algorithmic structure of the learned model \[1\]. This offers a potential path to rigorously defining empirically observed phenomena like “circuits” in models, in which case the loss kernel, by virtue of its connection to SLT, would be interpreted as a tractable empirical probe for these objects.

**Further experiments**
Several reviewers requested additional experiments, including requests for additional comparisons, additional quantitative metrics, or additional experimental settings.

We appreciate the reviewers’ suggestions and will consider incorporating them into future work. With this being said, while we believe the work would be strengthened by further experiments, we believe the existing work is sufficient to establish the validity and usefulness of the method.

We emphasize that many additional experiments are presented in the appendix. For instance, several reviewers asked for a demonstration of potential applications; we present a memorization detection experiment in Appendix D.3. Reviewers requested quantitative evaluation beyond UMAPs; we actually performed several, including “Taxonomic Lift” detailed in Appendix D.2. In future versions of this work we will aim to make these contributions clearer in the main body of the paper, and consider expanding upon them further.

We again thank the reviewers for their time in reviewing our work. While we have not responded to all the reviewers’ individual comments, we have read them carefully and will address them in future submissions. We believe the work will be improved by this process.

\[1\] Murfet & Troiani “Programs as Singularities” 2025 arXiv:2504.08075

**Withdrawal Confirmation:**

I have read and agree with the venue's withdrawal policy on behalf of myself and my co-authors.